# Molecular Markers of Adult Neurogenesis in the Telencephalon and Tectum of Rainbow Trout, *Oncorhynchus mykiss*

**DOI:** 10.3390/ijms23031188

**Published:** 2022-01-21

**Authors:** Evgeniya V. Pushchina, Anatoly A. Varaksin, Dmitry K. Obukhov

**Affiliations:** 1A.V. Zhirmunsky National Scientific Center of Marine Biology, Far Eastern Branch, Russian Academy of Sciences, 690041 Vladivostok, Russia; anvaraksin@mail.ru; 2Department of Biology, Kafedra Cytology and Histology, St. Petersburg State University, 199034 Saint-Petesburg, Russia; dkobukhov@yandex.ru

**Keywords:** glutamine synthetase, adult neural stem cells, radial glia, neuroepithelial cells, vimentin, nestin, doublecortin, adult neurogenesis, trout

## Abstract

In the brain of teleost fish, radial glial cells are the major type of astroglial cells. To answer the question as to how radial glia structures adapt to the continuous growth of the brain, which is characteristic of salmonids, it is necessary to study various types of cells (neuronal precursors, astroglial cells, and cells in a state of neuronal differentiation) in the major integrative centers of the salmon brain (telencephalon and *tectum opticum*), using rainbow trout, *Oncorhynchus mykiss*, as a model. A study of the distribution of several molecular markers in the telencephalon and tectum with the identification of neural stem/progenitor cells, neuroblasts, and radial glia was carried out on juvenile (three-year-old) *O. mykiss*. The presence of all of these cell types provides specific conditions for the adult neurogenesis processes in the trout telencephalon and tectum. The distribution of glutamine synthetase, a molecular marker of neural stem cells, in the trout telencephalon revealed a large population of radial glia (RG) corresponding to adult-type neural stem cells (NSCs). RG dominated the pallial region of the telencephalon, while, in the subpallial region, RG was found in the lateral and ventral zones. In the optic tectum, RG fibers were widespread and localized both in the marginal layer and in the periventricular gray layer. Doublecortin (DC) immunolabeling revealed a large population of neuroblasts formed in the postembryonic period, which is indicative of intense adult neurogenesis in the trout brain. The pallial and subpallial regions of the telencephalon contained numerous DC+ cells and their clusters. In the tectum, DC+ cells were found not only in the *stratum griseum periventriculare* (SGP) and longitudinal torus (TL) containing proliferating cells, but also in the layers containing differentiated neurons: the central gray layer, the periventricular gray and white layers, and the superficial white layer. A study of the localization patterns of vimentin and nestin in the trout telencephalon and tectum showed the presence of neuroepithelial neural stem cells (eNSCs) and ependymoglial cells in the periventricular matrix zones of the brain. The presence of vimentin and nestin in the functionally heterogeneous cell types of adult trout indicates new functional properties of these proteins and their heterogeneous involvement in intracellular motility and adult neurogenesis. Investigation into the later stages of neuronal development in various regions of the fish brain can substantially elucidate the major mechanisms of adult neurogenesis, but it can also contribute to understanding the patterns of formation of certain brain regions and the involvement of RG in the construction of the definite brain structure.

## 1. Introduction

Studies of adult animal neurogenesis are an important area of developmental neurobiology [1]. The interest in these studies is explained by the necessity to identify different types of cells with stem properties in the adult animal brain for the further assessment of the heterogeneity of neurogenesis, which is manifested both at the level of individual constitutive neurogenic niches and between different species [2,3,4].

In the brain of a teleost fish, radial glial cells are the main type of astroglial cells. To answer the question as to how the radial glia structures adapt to the continuous growth of the brain characteristic of salmon fish, it is necessary to study various types of cells (neuronal precursors, astroglial cells, and cells in a state of neuronal differentiation) in the main integrative centers of the brain: telencephalon, *tectum opticum*, cerebellum and brainstem of rainbow trout, and *Oncorhynchus mykiss*. Most teleost fishes continue to grow after reaching sexual maturity, which entails an increase in the size and in the number of cells of the brain, as has been shown for various teleost fish species [5,6,7,8]. The rainbow trout *Oncorhynchus mykiss* sometimes reaches a body length of 1 m and a body weight of up to 20 kg, but in most cases, trout is 20–30 cm long and weighs 400–600 g. However, the brain size in this fish species increases significantly with growth, more than 100-fold in volume. Various aspects of constitutive neurogenesis, such as environmental exposure [9], the influence of circadian rhythms [10], and sexual dimorphism [11], have been investigated. Neurons and glia are added to the brain by various ways in the process of constitutive growth: the appearances of neurons from a separate proliferative zone, e.g., the optic tectum, cerebellum, brainstem, and diencephalon; and the generation of new cells from radial glia [6,7,12].

The adult fish brain is a zone of continuous intense proliferative activity, not limited only to the telencephalon [7,13]. In some areas of proliferation in the brain, tissue-specific stem cells have neuroepithelial (NE), rather than radial glial (RG), characteristics [14,15]. Glutamine synthetase is a molecular marker of NSC, RG and progenitor cells [5,7]. Studies on juvenile salmonids have shown that, in the telencephalon of the masu salmon *Oncorhynchus masou* [16] and the chum salmon *Oncorhynchus keta* [8], and in the chum salmon mesencephalic tegmentum [17], most of adult neural stem cells (aNSCs) have a neuroepithelial (NE) or radial-glial (RG) phenotype. Such cells divide symmetrically, constantly increasing the pool of neural precursors, which subsequently differentiate into neurons, glia, and ependymal cells, forming the structure of the central nervous system (CNS) [18]. Nestin and vimentin are molecular markers of progenitor cells and radial glia that exhibit the capability of self-renewal and multipotency [14,17]. Doublecortin (DC) is a microtubulin-associated protein used as a molecular marker of neuroblasts required for the migration of immature neurons in the vertebrate brain; the expression of DC is associated with neuronal differentiation and migration [3,5,7]. Of particular importance in the adult constitutive development of the salmonid CNS is the fetalization process associated with developmental delay and the retention of features of the embryonic brain organization [19]. Studies on the medaka, *Oryzias latipes*, and the zebrafish, *Danio rerio*, have also shown that, within proliferative populations in the optic tectum, aNSCs are indeed NE-like stem/progenitor cells, but not RG cells [20,21]. Similar results were obtained in other studies, in particular, on the optic tectum [22] and telencephalon [23]. The study of the lateral zebrafish pallium revealed a population of adult NE cells [24,25] maintained from an early to various postembryonic stages of ontogenesis. All of these data emphasize the importance of non-RG stem cells in the neurogenesis of adult animals, such as fish. However, the distribution, cell-cycle characteristics, and molecular markers of NE cells and glial progenitors in fish at the adult stages are still poorly understood.

Recent studies of the telencephalon structure in *Astatotilapia burtoni*, in both small and large fish, have shown the presence of an aberrant pattern of RG fibers in the central pallium [26]. The main glial processes in *A. burtoni* bend around the nuclei of the central pallium, especially in the posterior part of the telencephalon. The analysis of RG growth by using immunocytochemical markers of proliferation, such as PCNA, stem cells Sox2, and neuronal differentiation DC, were revealed [26]. In addition, DC- and Sox2-positive cells were found in the deeper nuclei of the central part of the pallium [26]. These data suggest that RG cells give rise to migrating cells that deliver new neurons and glia to deeper pallial regions, leading to expansion of the central regions of the pallium and displacement of RG patterns. Thus, studies on *A. burtoni* have shown that RG cells can adapt to the morphological growth processes in the brain of adult fish and contribute to this growth.

Investigation into the later stages of neuronal development in various parts of the fish brain can elucidate the major mechanisms of adult neurogenesis and contribute to understanding the patterns of building of certain brain regions and the involvement of RG in the construction of the definite brain structure [11,27]. To answer the question as to how the structures of RG and the population of NE cells adapt to the continuous growth of the brain, which is characteristic of salmonids, adult stem cells (i.e., cells in the process of neuronal differentiation) were studied in two integrative brain centers, namely the telencephalon and the optic tectum, of juvenile (three-year-old) *O. mykiss*.

## 2. Results

### 2.1. Glutamine Synthetase in the Telencephalon of Trout

In the trout telencephalon, the dorsal (D) and ventral (V) areas were identified that correspond to the pallial and subpallial regions of the zebrafish [28]. The dorsal (Dd), lateral (Dl), central (Dc), and medial (Dm) zones were distinguished within the pallial region. In the subpallial region, the dorsal (Vd) and ventral (Vv) areas and the lateral (Vl) zone were differentiated.

In the juvenile trout pallium, glutamine synthetase (GS) labeling was revealed in the cells of the periventricular zone (PVZ), forming extensive areas in the form of a monolayer, or including local aggregations of immunopositive cells (Figure 1A, red inset). GS labeling was detected in intensely labeled small NE cells of rounded or oval shape and devoid of processes (Figure 1A, red inset; and Table 1). Another type of GS positive (GS+) cell was represented by RG; the bodies of the RG cells were located in the PVZ, and their long processes penetrated into the subventricular zone (SVZ) and deep into the parenchymal layers of the telencephalon (Figure 1A, red inset; and Table 1). In the SVZ, single GS labeled cells were identified; in the deeper parenchymal layers (PZ), single moderately labeled GS cells were found (Figure 1A and Table 1).

The Dc is characterized by a complex pattern of GS distribution (Figure 1B and Table 1). In this area of the pallium, three types, moderately and intensely labeled cells of small size (Table 1), and also two types of immunopositive fibers were identified. RG was represented by moderately labeled fibers of the first type, extending from the periventricular areas of the pallium (Figure 1B, black inset). Intensely labeled fibers of the second type formed varicose thickenings and were glutamatergic fibers of varying degrees of maturity (Figure 1B, black inset). Intensely labeled terminals of glutamatergic fibers forming simple terminal extensions or clavate-shaped terminals were frequently detected on bodies of moderately GS-labeled cells in the Dc (Figure 1B, black inset). More complex intensely labeled terminal apparatuses of glutamatergic fibers enclosing GS cells were rarely encountered (Figure 1B, red inset). The number of GS+ cells in Dc was minimal, significantly differing from the Dd (*p* < 0.01) and Dm (*p* < 0.05) (Figure 1M). The density of RG distribution in the Dc was also lower than in other areas of the pallium (Figure 1N).

In the Dd, small intensely labeled GS+ cells of the NE type in the PVZ were located above the layer of immunonegative cells (Figure 1C and Table 1). In the SVZ, there were single cells with moderate GS labeling, sometimes forming small clusters (Figure 1C, yellow arrows; and Table 1). In the deeper layers (PZ), heterogeneous cell complexes were identified, including immunonegative cells adjoining clusters of small moderately labeled GS+ cells and/or nerve terminals (Figure 1C, inset). In the Dd, the number of GS+ cells was the maximum (Figure 1M).

In the pallium of juvenile (three-year-old) trout, RG was distributed in the medial, dorsal, and lateral regions, forming guides for the migration of mature cells (Figure 1D, inset). The Dm was characterized by the presence of intensely labeled NE cells arranged into small groups alternating with small immunonegative areas (Figure 1F and Table 1). Single moderately or weakly labeled cells were observed in the SVZ (Figure 1F). In the deeper (PZ) layers of the Dm, RG fibers and also intensely labeled terminals of glutamatergic fibers (Figure 1F, inset), converging on single or paired immunonegative cells, were identified (Figure 1F, inset). In the Dm, single intensely labeled glutamatergic fibers with varicose microsculptures were identified (Figure 1F, inset). The distribution density of GS+ cells was lower than in the Dd (Figure 1M), and the radial glia was maximum among all the pallial zones (Figure 1N).

At the border between the dorsal and medial pallial zones, a change in the direction and density of the RG distribution was clearly observed (Figure 1F). In the border area, the distribution density of cells and their morphological characteristics also changed, making it possible to accurately identify the boundaries of the two pallial zones (Figure 1F). In the area of the PVZ in the border area, no morphological features were revealed, and the general pattern of distribution of GS+ neuroepithelial cells and RG cells remained unchanged (Figure 1F). In the ventral part of the border zone, an area of increased distribution density of intensely labeled GS+ terminals was identified (Figure 1F, inset).

At the border between the dorsal (Dd) and lateral (Dl) zones of the pallium, a similar change in the density of glutamatergic innervation was observed (Figure 1G, inset). The PVZ of the lateral region was characterized by a clustered pattern of distribution of GS+ cells of the NE and RG types (Figure 1G). In the SVZ and deeper layers of the PZ, it was possible to trace bundles of RG fibers extending from the constitutive domains (Figure 1G). On the Dd side, an increased density of distribution of GS+ fibers and their terminals on the bodies of immunonegative cells was revealed (Figure 1G, inset). In the region of the Dd/Dl transition, larger GS cells were identified, with dense and large GS+ terminals converged on their bodies (Figure 1G, inset). Thus, areas of increased density of distribution of glutamatergic innervation were revealed at the Dm/Dd and Dd/Dl boundaries of the pallial zones.

In the lateral zone of the pallium, a high density of RG distribution was determined (Figure 1H). GS+ cells in the PVZ formed a dense monolayer, with the RG located in the basal part and NE cells in the apical part (Figure 1H, inset). The migration patterns of GS cells along GS+ RG fibers were identified at a considerable distance deep into the brain (Figure 1H). The distribution density of GS+ cells in the Dl was significantly lower (*p* < 0.05) than in the Dd (Figure 1M).

In the subpallial region of the trout telencephalon, GS was found in the intensely labeled dorsal Vd and ventral Vv cell groups (Figure 1I, inset; and Table 1), as well as in the RG fibers of the lateral Vl regions (Figure 1I and Table 1). Intense GS labeling was identified in NE cells in the apical part of the PVZ and in the loci of intensely labeled innervation in the SVZ (Figure 1I, inset). Densely labeled GS+ cells were found in the periventricular Vd cell masses; an intensely labeled neuropil was identified in the SVZ (Figure 1J, inset).

In the Vl, single small intensely labeled aggregations of NE cells in the PVZ were visualized; in the deeper layers of the SVZ and PZ, labeled RG fibers were identified, along with immunonegative cells (Figure 1K, inset; and Table 1). At the border between the lateral pallium (Dl) and subpallium (Vl), an increased density of distribution of RG and GS+ innervation was found along an aggregation of immunonegative cells (Figure 1L, inset).

### 2.2. Glutamine Synthetase in the Trout Tectum Opticum

In the *tectum opticum* of trout, GS distribution was found in the cells of the marginal layer (SM) and periventricular gray layers (SGP) (Figure 2A and Table 1). In the SM and SGP, two types of NE cells (undifferentiated small and oval cells) and two types of RG cells were identified (Table 1). A general view of GS immunolabeling in the tectum is shown in Figure 2A. GS+ neuroepithelial and RG progenitors were found in both SM (Figure 2A, blue inset) and SGP (Figure 2A, red inset).

In the dorsal part of the tectum in the SM, four types of intensely labeled GS+ cells were identified, with two types of NE cells distinguished among them: small rounded and larger elongated ones, devoid of processes (Figure 2B and Table 1). NE cells of both types, as a rule, were located singly, or formed small constitutive clusters located in the apical zone of the SM (Figure 2B, red inset). Under the neuroepithelial GS+ cells, clusters of larger immunopositive RG cells were observed, whose processes extended into the deeper layers of the tectum (Figure 2B, blue inset). In the dorsal part of the tectum, two types of RG cells were identified, differing in the size of their bodies (Table 1). RG cells often formed GS+ clusters separated by immunonegative zones (Figure 2B, blue inset). The RG branches differed in thickness and length: the RG type 2 had thicker and longer branches compared to the branches of the RG type 1 (Figure 2B). In the *stratum griseum centrale* (SGC) and the *stratum griseum et album periventriculare* (SGAP), an increased density of GS+ fibers was revealed, with varicose thickenings forming areas of increased glutamatergic innervation observed along them (Figure 2B, in white rectangle). In the SGP region, two types of intensely labeled GS+ cells were identified: elongated NE cells and type-1 RG cells (Figure 2C and Table 1). These cells formed mixed-type clusters, including GS+ NE cells in the apical zone and GS+ RG in the basal zone, surrounded by GS-negative SGP cells (Figure 2C, inset). Densely located homogeneous aggregations of immunopositive cells were distinguished in the composition of the clusters (Figure 2C).

In the dorsolateral part of the tectum in the area of the SGP, fibers of the lateral optical tract (LOT) were localized, which penetrated the large vessels of the intertectal vascular plexus (Figure 2D). In this area of the tectum, a maximum density of GS+ innervation in SGC and SGAP was revealed (Figure 2D, white rectangle). In the lateral part of the SM tectum, three types of GS+ cells were identified: NE cells of types 1 and 2, and type-1 RG cells (Table 1). These cells formed neurogenic clusters whose density of distribution and length were lower than in the dorsal tectum (Figure 2D, inset). In contrast, four types of intensely labeled GS+ cells were identified in SGP (Table 1).

Topographically, the localization of the heterogeneous population of GS+ NE cells and RG cells in the lateral tectum was highly specific and formed a large latero-caudal constitutive proliferative tectal zone (Figure 2F). Constitutive neurogenic niches with complex compositions (Figure 2F, inset), including the major part of NE cells and a smaller part of RG cells (Figure 2F, inset), were revealed in the lateral periventricular proliferative tectal zone. In the lateral proliferative zone of the tectum, a high density of glutamatergic innervation with varicose thickenings was also observed (Figure 2F, white arrowheads). The periventricular GS+ cell populations that formed neurogenic constitutive neurogenic niches in the lateral proliferative tectal zone also had a complex cellular composition, including NE and RG cells (Figure 2F). Thus, in the lateral proliferative zone of the tectum, the periventricular zone (PVZ, SGP) was more developed than in the dorsal tectal zone and prevailed over the SM (Figure 2G). It contained both homogeneous areas, including NE cells (Figure 2H, white rectangle), and hypertrophied zones, including heterogeneous GS+ cell clusters (Figure 2H, red rectangle). Such areas had a somatotopic principle of organization, were characterized by a complex cell composition, and exhibited a high neurogenic activity (Figure 2I). The results of a comparative analysis of the distribution of GS+ cells in the lateral and dorsal tectum showed that in both regions the amount of RG in the *stratum marginale* (SM) is greater than in the PVZ (Figure 2J); however, no significant intergroup differences were found. In contrast, the number of GS+ NE cells showed significant intergroup differences in PVZ (*p* < 0.01) and MS (*p* < 0.05) (Figure 2K). Thus, the number of NE cells in the lateral PVZ of the tectum is multifold greater than in the dorsal (*p* < 0.01), while in MS, on the contrary, it is significantly reduced (*p* < 0.05). In the PVZ, significant intergroup differences (*p* < 0.05) were found in the number of cells in the lateral and dorsal zones of the tectum (Figure 2L). In the MS, the total number of GS+ cells in the dorsal part was, on the contrary, significantly larger (*p* < 0.05) than the number in the lateral one (Figure 2M).

### 2.3. Doublecortin in the Trout Telencephalon

As a result of doublecortin immunolabeling in trout, DC+ cells in all areas of the telencephalon were detected both in the periventricular and subventricular zones and in deeper parenchymal areas (Figure 3A and Table 2). The intensity of cell immunolabeling in the Dd varied from intense to moderate: in the intense case, the cell body was completely labeled, while in the moderate case, the cytoplasm was stained and the nucleus remained immunonegative (Figure 3A, inset). Cell immunolabeling patterns were usually uniform or granular (inset Figure 3A,B). DC+ cells were detected in all areas of the pallium and subpallium, within diffuse parenchymal aggregates (Figure 3B, inset; and Table 2) or as denser conglomerates located in the SVZ (Figure 3B).

Moderately labeled polygonal or elongated cells dominated the diffuse Dd conglomerates (Figure 3C, inset; and Table 2). In denser conglomerates, intensely labeled cells were located in the apical part of the SVZ, while less intensely labeled cells spread to the basal part (Figure 3C). Several types of DC+ cells at different stages of growth and differentiation, varying in size and topography, were identified (Table 2). The first type of cells was represented by small rounded undifferentiated cells located in the PVZ in the immediate vicinity of the larger elongated cells in the SVZ (Figure 3D and Table 2). Dense aggregations of DC+ cells of varying degrees of differentiation, located directly in the PVZ and SVZ, corresponded to constitutive neurogenic niches. Larger oval cells were found in the area of the SVZ and in the deeper layers of the PZ; they were intensely or moderately DC-immunopositive and belonged to type 3 (Table 2). Cells of a larger size, polygonal shape, moderately DC-immunopositive, and usually found in PZ belonged to types 4 and 5 (Table 2).

In the central zone of the pallium Dc, larger cells of types 3–5 were revealed (Figure 3F and Table 2). The intensity of DC+ cell immunolabeling varied from intense to moderate (Figure 3F and Table 2). Intensely labeled cells often had two to four processes (Figure 3F). The density of distribution of cells was relatively high; in the sparser areas of Dc, long and sometimes branched processes of DC+ cells were traced (Figure 3G). At the border between the central Dc and lateral Dl pallial zones, the number of intensely labeled DC cells increased (Figure 3H). In the Dl area, in the border zone, small intensely DC labeled type-2 cells were encountered, and RG fibers were also traced (Figure 3H inset and Table 2). In the Dl, four types of DC+ cells were identified (Figure 3I and Table 2). In the PVZ and SVZ, intensely or moderately labeled cells dominated, forming dense clusters (Figure 3I,J). RG fibers, larger polygonal cells of types 1 and 2 with moderate or intense DC labeling, forming clusters of a diffuse type, were identified in the deeper layers of the PZ (Figure 3I,J). In the Dm, the distribution density of DC+ cells was higher than in other pallial areas; this was especially evident at the border between the Dm and Dc (Figure 3K). In the Dm, four types of intensely or moderately DC+ cells were identified, whose sizes were smaller than in other areas of the pallium (Figure 3L and Table 2).

In the trout subpallium, DC immunolocalization was detected in the dorsal (Vd), ventral (Vv), central (Vc), and lateral (Vl) regions (Figure 3M). In the lateral (Vl) zone, where the intensity of DC immunolabeling varied from intense to moderate, four types of cells were identified (Figure 3M, inset; and Table 2). A noticeable thickening of the periventricular Vd and Vv regions formed a hypertrophied cellular structure, in which the central (Vc) zone was also identified, where small intensely labeled and larger moderately DC+ cells were encountered (Figure 3N, inset; and Table 2). In the Vd, a high density of distribution of three types of DC+ cells was observed in the PVZ, SVZ, and PZ (Figure 3O and Table 2). In the Vv, intensely and moderately DC-immunopositive cells of three types dominated the PVZ and SVZ (Figure 3R and Table 2); in the PZ, a stratified pattern of DC+ cell distribution was revealed (Figure 3R).

Thus, in the pallial region of the trout telencephalon, the patterns of doublecortin distribution varied between different regions in terms of the morphological composition of DC+ cells, intensity of immunolabeling, spatial aggregation, and the relationship with RG. The results of a quantitative analysis of the distribution of DC+ cells are shown in Figure 3Q. The maximum number of DC+ cells was found in the medial part of the pallium, and the minimum was in the lateral part (Figure 3Q). In the dorsal and central zones, the number of DC+ cells was relatively high (Figure 3Q). In the subpallial region, a complex pattern of DC immunopositivity was also present in various zones, including the areas of increased density of DC immunolabeled cells in the dorsal zone and a stratified structure in the ventral zone, thus indicating an intensive migration of neuroblasts formed during the constitutive neurogenesis. In the subpallial region, the numbers of DC+ cells in the ventral and lateral zones were approximately similar and slightly decreased in the dorsal zone (Figure 3Q).

The one-way analysis of variance (ANOVA) of the comparative distribution of GS+ RG and DC+ cells in the pallial zones of trout showed that the number of GS+ RG cells was many times lower than the number of DC+ cells (intergroup differences *p* < 0.01) in the dorsal, medial, and central zones of the pallium and significantly lower (intergroup differences *p* < 0.05) in the lateral zone (Figure 3R). An analysis of the ratio of GS+ NE cells to the number of DC+ cells showed intergroup differences for the lateral (*p* < 0.05), medial, and central zones (*p* < 0.01) and did not reveal intergroup differences for the dorsal zone (Figure 3S). Thus, in the dorsal zone, the number of NE progenitors did not significantly differ from the number of DC+ cells found in this zone, and this may indicate a high neurogenic potential of this area of the pallium as compared to other areas. As a result of increase in the number of GS+ progenitors of embryonic (NE) and adult (RG) types, we obtained a total value of the number of GS+ progenitors in the pallium, which was compared with the numbers of DC+ cells in all regions (Figure 3T). The results of one-way ANOVA showed significant differences in the ratio of the total numbers of GS+ and DC+ cells between the medial and central zones (*p* < 0.05) and between the lateral and dorsal zones; the total number of GS+ cells did not significantly differ from the number of DC+ cells formed in the postembryonic period.

### 2.4. Doublecortin in the Trout Tectum Opticum

In the trout *tectum opticum*, the doublecortin immunolocalization was detected in different layers (Table 2). A general view of the DC distribution in the dorsolateral part of the tectum is shown in Figure 4A. In the MS, two types of small intensely labeled cells and RG cells were identified (Figure 4A, inset; and Table 2). In the layer of *stratum opticum* (SO) optic fibers, two types of intensely labeled DC+ cells were revealed (Figure 4B, inset; and Table 2). Five types of intensely and moderately labeled DC cells were detected in the SGC (Figure 4B and Table 2). Small type-1 and -2 cells were intensely labeled; in larger cells of types 3–5, the cytoplasm was intensely labeled, and the nucleus was immunonegative (Figure 4B). The presence of extensive constitutive neurogenic niches located in the basal part of the MS and/or SO was a feature of the dorsolateral tectum of trout (Figure 4C, white dotted line). Intensely labeled RG fibers approached such niches from the MS side (Figure 4C, inset).

In the lateral tectum, the patterns of DC+ cell distribution were similar, with the exception of SGP, where an increased number of intensely labeled immunopositive cells was found (Figure 4D, inset). In the area of the lateral periventricular matrix zone of the tectum, undifferentiated DC+ cells were found not only in the SGP; a migrating population of larger undifferentiated DC+ cells was found within the periventricular gray and white layers of SGAP (Figure 4D, inset; and Table 2). The cell composition of SGAP was similar to that of SGP (Table 2). In the lateral tectum, even larger constitutive niches were found in the MS and SO (Figure 4E). The largest single bipolar DC+ cells with an immunonegative nucleus were found in the SGC of the lateral tectum (Figure 4F, inset). In the SGPs of both lateral (Figure 4F, inset) and dorsolateral tectum (Figure 4G, blue inset), a dense layer of intensely labeled DC+ cells and patterns of DC migration of immunopositive cells were observed. In the dorsolateral zone of the tectum, in the MS and SO, a network of RG fibers was found (Figure 4G, black inset).

In the paramedian region of the tectum, a heterogeneous population of DC+ cells was identified as part of the *torus longitudinalis* (TL) (Figure 4H, inset; and Table 2). In the TL, four types of intensely and moderately DC+ cells were distinguished (Table 2). An extensive aggregation of elongated and polygonal moderately DC+ labeled cells with immunonegative nuclei was localized ventrally of the *intertoral commissure* (CT) (Figure 4H, inset). Small intensely and moderately labeled type-1 and -2 cells, as well as RG fibers with neuroblasts migrating along them, were localized in the periventricular and subventricular layers of the TL (Figure 4H). The CT fibers were a continuation of the SGAP of the medial tectum; the heterogeneous population of DC+ cells, located directly below the CT, contained larger polygonal immunopositive cells (Figure 4I, inset). Thus, DC-expressing neuroblasts and migrating DC+ cells were identified in different parts of the optic tectum of trout; their quantitative ratio in different layers of the dorsal and lateral tectum is shown in Figure 4J. The maximum number of DC+ cells was found in SGP, and the minimum in SO (Figure 4J). In the area of the dorsolateral and lateral tectum, extensive constitutive neurogenic zones were identified, thus indicating an intense constitutive neurogenesis in the tectum, which is an integrative center of the trout brain.

The ratio of DC+ cells in SGP and DC+ RG in MS in the lateral and dorsal tectum was estimated using one-way ANOVA; as a result, significant (*p* < 0.01) intergroup differences were revealed in both zones, thus indicating the dominance of NE cells over the RG in the dorsal and lateral tectum of trout (Figure 4K). The estimation of the ratio of GS+ progenitors in SGP and GS+ RG in MS in the lateral and dorsal tectum showed significant (*p* < 0.05) intergroup differences in the lateral tectum, thus indicating the dominance of GS+ RG in the lateral tectum (Figure 4K). The total number of GS+ progenitors of embryonic (NE) and adult (RG) types in the dorsal and lateral tectum was compared with the number of DC+ cells in the same areas (Figure 4L). The one-way ANOVA showed significant differences in the ratio of the total number of DC+ and GS+ cells in the lateral tectum (*p* < 0.01), and also significant differences (*p* < 0.05) in the same parameters in the dorsal tectum, thus indicating the dominance of DC+ cells formed in the postembryonic period over the number of GS+ progenitors of embryonic (NE) and adult (RG) types.

### 2.5. Vimentin in the Trout Telencephalon

In the juvenile trout pallium, intensive vimentin labeling was detected in PVZ cells forming an extensive monolayer or including local aggregations of immunopositive cells (Figure 5A,B, insets). The vimentin labeling of in the Dd was detected in small NE cells, intensely labeled rounded cells, or oval cells devoid of processes (Figure 5B, inset; and Table 3). Vimentin-immunopositive (Vim+) NE cells formed constitutive clusters in the basal part of the PVZ (Figure 5B, white ovals). Another type of Vim+ Dd cells was represented by RG; the bodies of RG cells were located in the PVZ, and their short processes penetrated into the SVZ of the telencephalon (Figure 5A, inset; and Table 3).

A high density of distribution of NE cells in the PVZ was observed in the pallium lateral zone (Dl) (Figure 5C and Table 3). Vim+ cells in the basal part of the PVZ formed dense isolated groups that were sometimes sparser and more extensive clusters (Figure 5C, inset; and Table 3). In some areas of the Dl, areas containing RG bundles were identified (Figure 5D, inset). The processes of RG cells in such groups extended to the SVZ, and the cell bodies were organized into dense compact clusters (Figure 5D, inset). The medial area of the pallium contained Vim+ NE type cells in the PVZ, forming a monolayer (Figure 5F, inset; and Table 3). The intensity of cell immunolabeling was very high; in the Dm, the smallest undifferentiated Vim+ cells were detected, while RG was not found (Table 3). At the border between the dorsal and medial areas of the pallium, the distribution density of Vim+ cells was reduced (Figure 5F, inset). On the Dd side, more noticeable constitutive Vim+ clusters of NE cells prevailed; on the Dm side, the Vim+ NE cell distribution pattern in the border area corresponded to a sparse monolayer (Figure 5F, inset).

In the subpallium region, a very low immunopositivity to Vim was detected in the dorsal region of Vd (Figure 5G, inset; and Table 3). Several Vim+ cells of the NE type were identified in the PVZ. Diffusely organized weakly labeled cells were identified in the SVZ and PZ (Figure 5G, inset; and Table 3). At the border between the Dl and Vl, Vim+ cells were almost absent; on the Dl side, clusters of immunopositive cells were observed in the PVZ (Figure 5I, black inset). On the Vl side, a small single aggregation of Vim+ NE type was detected (Figure 5I, red inset; and Table 3). Thus, in the subpallial region, the distribution of Vim+ cells of the NE type was very limited, and RG was not detected. The analysis of the quantitative distribution of Vim+ cells of NE type and RG is shown in Figure 5J. The ANOVA test showed intergroup differences in the distribution of Vim+ precursors of NE type and RG in the dorsal and lateral zones of the pallium (*p* < 0.05). The data of a quantitative analysis showed the maximum number of Vim+ cells of the NE type in the medial zone (Figure 5J). The analysis of the ratio of the amount of Vim+ and GS+ RG cells in the pallial area of trout is shown in Figure 5K. The ANOVA test showed intergroup differences (*p* < 0.05) in the distribution of Vim+ and GS+ RG cells in the dorsal and lateral zones (Figure 5K). This indicates the dominance of GS+ RG, which corresponds to adult progenitors in the dorsal and lateral pallium. The analysis of the ratio of the number of Vim+ and GS+ cells of the NE type in the pallial area of trout is shown in Figure 5L. The ANOVA test showed intergroup differences (*p* < 0.05) in the distribution of Vim+ and GS+ cells of the NE type in the dorsal zone, indicating the dominance of GS+ progenitors of the embryonic type in the Dd. In the lateral and medial pallium, no significant intergroup differences were found, thus indicating similar proportions of Vim+ and GS+ cells of the NE type in the Dm and Dl. 

### 2.6. Vimentin in the Trout Tectum

In the trout optic tectum, a low level of immunopositivity of vimentin was revealed (Figure 6A). However, five types of Vim+ NE cells were identified in SM; neurogenic niches of various sizes were found in ependyma above SM (Figure 6B, inset; and Table 3). Part of the Vim+ cell population in the tectum was represented by elongated intensely or moderately labeled cells in a state of migration (Figure 6C, inset; and Table 3). Immunonegative cavities surrounded by numerous elongated immunonegative cells, which corresponded to lumens of large vessels, were often found in the lateral tectum in the SGP (Figure 6D, inset; and yellow arrow). In the SGP of the dorsolateral part of the tectum, there were few, single Vim+ type-3 and -4 cells (Figure 6E, inset; and Table 3). In the basal part of the SGP, a high density of distribution of elongated immunonegative cells was revealed (Figure 6E, inset, yellow arrow). Laterally, there was a SGP thickening including diffuse patterns of distribution of small intensely labeled Vim+ cells (Figure 6F, inset, red arrow), co-localized with larger weakly or moderately labeled cells (Figure 6F, inset, white arrow). In the dorsal part of the tectum, the SGP was dominated by larger intensely or moderately labeled type-4 and -5 cells (Figure 6G, inset; and Table 3). In the area of the lateral proliferative zone of the tectum in the SGP, single intensely Vim+ labeled cells of the NE type and complex morphological patterns of resident and migrating immunonegative cells were revealed (Figure 6H, inset). The ependyma contained numerous immunonegative cells and dense conglomerates including moderately or intensely labeled Vim+ cells (Figure 6I, inset).

The results of a quantitative analysis of distribution of Vim+ cells of NE type in the PVZ and SM of the dorsal and lateral tectum are shown in Figure 6J. The one-way ANOVA revealed intergroup differences (*p* < 0.05) in the distribution of immunopositive cells in the PVZ and MS. According to them, the number of Vim+ NE type cells in the PVZ of the lateral and dorsal tectum was significantly greater than that in the MS.

The analysis of the ratio of the numbers of Vim+ and GS+ cells of the NE type in the lateral and dorsal tectum of trout is shown in Figure 6K. The ANOVA showed intergroup differences (*p* < 0.05) in the distribution of Vim+ and GS+ cells of the NE type in the lateral tectum, indicating a dominance of Vim+ progenitors of the embryonic type over GS+ progenitors. In the dorsal tectum, on the contrary, the significant intergroup differences (*p* < 0.05) in the distribution of Vim+ and GS+ cells of the NE type indicated the dominance of GS+ progenitors over Vim+ progenitors of the embryonic type.

An estimation of the ratio of the numbers of Vim+ and GS+ precursors of the NE type in the PVZ and MS of the lateral and dorsal tectum by the one-way ANOVA showed that the total number of Vim+ and GS+ embryonic progenitors in the lateral proliferative zone was multifold (*p* < 0.001) higher than that in the dorsal PVZ (Figure 6L). No significant intergroup differences were found for the MS of the lateral and dorsal tectum (Figure 6L).

### 2.7. Nestin in the Trout Telencephalon

As a result of nestin immunolabeling, Nes+ cells were detected both in the PVZ and SVZ and in the deeper PZ in all areas of the trout telencephalon (Figure 7A and Table 4). Nes+ cells were found in all areas of the pallium and subpallium, within diffuse parenchymal aggregations (Figure 7A,B, inset; and Table 4). The intensity of immunolabeling of cells in the dorsal pallium (Dd) varied from intense to moderate: intensely labeled cells dominated the PVZ; moderately labeled cells with granular cytoplasmic localization of nestin were detected in the PVZ; the nucleus remained immunonegative (Figure 7B, inset; and Table 4). In the medial zone of the pallium, intensely labeled Nes+ cells were detected in the PVZ, where they formed a multistratum layer that included dense conglomerates of immunopositive cells; in addition, Nes+ cells were located in a single layer in the basal part of the PVZ that also included constitutive aggregations of intensely labeled cells (Figure 7C and Table 4). Cells with immunolabeled cytoplasm were localized in the SVZ (Figure 7C, red inset; and Table 4). In the PZ, there were single intensely labeled cells, as well as moderately labeled cells, with granular immunopositive inclusions in the cytoplasm (Figure 7C, yellow inset; and Table 4). Densely labeled Nes+ cells having a horizontally elongated shape were found at the border between the Dd and Dm in the SVZ (Figure 7D, inset). Type-2 Nes+ cells varying in size and intensity of immunolabeling were detected in the central zone of pallium (Dc) (Figure 7E, inset; and Table 4). Four types of moderately labeled polygonal Nes+ cells (Figure 7F, yellow inset; and Table 4), and also intensely labeled polygonal cells with processes were identified at the border between the Dm and Dc (Figure 7F, red inset; and Table 4). At the border between the lateral and dorsal zones of the pallium in the SVZ and the superficial parenchymal layers, there were RG patterns and an increased concentration of densely labeled cells and cells with granular cytoplasmic localization of nestin (Figure 7G, inset). In the lateral zone of the pallium, the density of distribution of Nes+ cells was increased; four types of cells were distinguished (Figure 7H, inset; and Table 4). The maximum density of distribution of Nes+ cells was found, and larger immunopositive cells were identified in the posterolateral area of the pallium (Dlp) (Figure 7I, inset; and Table 4).

In the subpallium, Nes+ cells were found in the dorsal (Vd), ventral (Vv), and lateral (Vl) zones (Figure 7J, inset; and Table 4). In the Vd, two types of Nes+ cells were identified, forming small and extensive constitutive clusters of immunopositive cells in the basal part of the PVZ (Figure 7K and Table 4). In the Vv, larger type-3 cells were localized in the PZ (Figure 7L and Table 4). In the Vl type 4, Nes+ cells were localized in a dense layer in the PVZ, as well as in the PZ (Figure 7J, red inset; and Table 4).

An analysis of the quantitative distribution of Nes+ cells in the pallial and subpallial zones is shown in Figure 7M. The ANOVA test showed intergroup differences in the distribution of Nes+ progenitors of the NE type between the central zone and the dorsal, medial, and lateral zones of the pallium (*p* < 0.05); the maximum number of Nes+ cells of the NE type was found in the medial zone (Figure 7M). In the subpallium, there were significant intergroup differences in the distribution of Nes+ progenitors between the ventral zone and the dorsal and central zones (*p* < 0.01); the maximum number of Nes+ cells was in the dorsal zone (Figure 7M).

The two-way ANOVA of the ratio of the numbers of Nes+, DC, and GS+ cells in the pallial region of trout are shown in Figure 7N. The ANOVA revealed intergroup differences (*p* < 0.05) in the distribution of GS and DC in the medial and central zones, as well as DC and Nes in the central zone of the pallium (Figure 7N). This indicates the dominance of DC+ neurons in the medial and central pallium over NSC/NPC in trout.

The comparative analysis of the distribution of Nes+ and DC+ cells of the NE type in the pallium and subpallium is shown in Figure 7O. The ANOVA revealed significant intergroup differences in the distribution of Nes+ and DC+ cells of the NE type in the central zone of the pallium (*p* < 0.01) and the ventral zone of the subpallium (*p* < 0.001), indicating the dominance of DC+ neurons over Nes+ NSC/NCP. No significant intergroup differences were found in the dorsal, lateral, and medial pallium, as well as in the dorsal subpallium (Figure 7O).

### 2.8. Nestin in the Tectum

In the optic tectum, nestin immunolocalization was detected in different layers (Table 4). A general view of the distribution of Nes+ cells and RG cells in the dorsolateral part of the tectum is shown in Figure 8A. In the MS, two types of small intensely labeled cells and RG cells were identified (Figure 8B, blue inset, and Table 4). In the SGC, type-3 intensely and moderately labeled Nes+ cells were identified (Figure 8B, red inset; and Table 4). Elongated and oval type-1 and -2 cells were intensely labeled (Figure 8C, white inset; and Table 4); in larger type-3 cells, cytoplasm was intensely labeled, and the nucleus was immunonegative (Figure 8C, red inset; and Table 4). Thus, a high density of distribution of various types of Nes+ cells was found in the SGC (Figure 8D). Two types of Nes+ cells were identified in the SGP of different areas of the tectum (Table 4). Immunopositive cells in the dorsolateral tectum formed local clusters, dense layers, or were organized into diffuse clusters (Figure 8F, inset). The dorsal tectum was dominated by a diffuse pattern of distribution of Nes+ cells in the SGP (Figure 8F, inset). In the lateral tectum, extensive, densely labeled cell conglomerates in the SGP alternated with areas devoid of immunopositive cells (Figure 8G, inset). In some cases, intensely labeled cells were distributed diffusely in the basal part of the SGP (Figure 8H, inset). In the TL, three types of Nes+ cells were identified; type-1 and -2 cells were found singly and/or as part of constitutive neurogenic aggregations in the periventricular areas (Figure 8I, inset; and Table 4). Type-3 cells were found in the TL parenchyma (Figure 8I).

The results of a quantitative analysis of the distribution of Nes+ cells in the dorsal and lateral tectum are shown in Figure 8J. The one-way ANOVA revealed intergroup differences (*p* < 0.05) in the distribution of immunopositive cells in the PVZ, SGAC, and SM (Figure 8J). The results showed that the number of Nes+ cells in SGAC and SM of the dorsal tectum was significantly greater than that in the lateral tectum, while, in the PVZ of the lateral tectum, the number of Nes+ cells was greater than that in the dorsal one (Figure 8J).

An analysis of the proportions of the numbers of GS+, Vim+, and Nes+ NE cells in the lateral and dorsal tectum of trout is shown in Figure 8K. The two-way ANOVA showed significant intergroup differences (*p* < 0.01) in the distribution of GS+ and Nes+ cells and significant intergroup differences (*p* < 0.05) in the distribution of Vim+ and Nes+ cells in the SGP of the lateral tectum, indicating a dominance of GS+ progenitors of the embryonic type. In the dorsal tectum, on the contrary, significant intergroup differences (*p* < 0.01) were found in the distribution of Nes+ and GS+ cells and significant intergroup differences (*p* < 0.05) in the distribution of Vim+ and GS+ cells of the NE type. This indicates a prevalence of embryonic Nes+ progenitors in the dorsal tectum in trout.

The one-way ANOVA of the ratio of the numbers of Nes+ and GS+ precursors of the NE and RG types in the PVZ and MS of the lateral and dorsal tectum showed significant intergroup differences in the distribution of precursors of various cell phenotypes (Figure 8L). In the MS of the dorsal tectum, the number of Nes+ precursors of the NE type was significantly greater than that of GS+ cells (*p* < 0.05), while, for the precursors of the RG type, on the contrary, the number of GS+ cells was greater than (*p* < 0.05) that of Nes+ progenitors (Figure 8L). In the SGP of the lateral tectum, the number of NE type Nes+ progenitors was significantly greater than (*p* < 0.05) the number of GS+ NE progenitors (Figure 8L). For other phenotypes, no significant intergroup differences were found in the lateral and dorsal tectum (Figure 8L).

## 3. Discussion

Adult neurogenesis is involved in the long-term postnatal development of the brain. Persistent addition of new neurons (constitutive neurogenesis) or regeneration after damage is a widespread phenomenon among various animals [4]. Shortly after the Altman’s discovery of neurogenesis in adult mammals [29], only few species of the extensive superclass of teleost fishes have been studied, although the adult cell proliferation was considered in them [26,30,31]. The adult neurogenesis has been reported for even fewer teleosts. The processes of constitutive brain growth in trout are significantly influenced by the fetalization process associated with the embryonic traits in adult animals [19]. At the phenomenological level, it is present in a large number of NE-type precursors in the telencephalon and optic tectum. The functional role of this feature still remains unclear.

### 3.1. Telencephalon

#### 3.1.1. Expression of Glutamine Synthetase in the Telencephalon

The patterns of GS distribution in the *O. mykiss* telencephalon indicate a dense arrangement of RG cells at the edge of the brain corresponding to the surface of the everted cerebral ventricle. However, intensely labeled GS+ cells devoid of processes were observed in the central zone (Dc) of the pallium (Figure 1A). Similar clusters of GS+ cells in similar areas of the pallium were reported for *Astatotilapia burtoni* [26]. The major glial processes in *A. burtoni* were found bending around the nuclei of the central pallium, especially in the posterior part of the telencephalon. The distribution patterns of GS+ RG in trout generally resemble those in large individuals of *A. burtoni*; however, using GS immunoperoxidase labeling, we revealed some additional features that are not characteristic of other studied fish species. Among them are the terminal apparatuses intensely immunolabeled with GS that braid the surfaces of immunonegative cells and represent the terminal targets of glutamatergic innervation (Figure 1B, red inset). Glutamate, along with its function of an excitatory neurotransmitter, is considered a mediator of neural inflammation associated with ischemic brain pathology or traumatic brain injury [32,33]. During the constitutive neurogenesis, most of newly formed neurons in the fish brain die within the first four weeks after intense proliferation [7,34]. Their selection is regulated by chemical signals, among which glutamate plays an important role [4,5]. The sections through the trout telencephalon showed that neuroblasts in the parenchymal zone of the brain express an “unequal set” of glutamate receptors, having the form of terminal thickenings of various shapes and sizes on the cell surface, as well as fibers (Figure 1C, inset). The studies of neurospheres obtained from cells of the subventricular zone of the mammalian telencephalon showed that agonists of mGluR2 receptors stimulate cell proliferation, while simultaneously reducing the number of apoptosis markers [35]. Signals from glutamate inhibit the proliferation of undifferentiated neural progenitors. However, the expression of III mGluR glutamate receptors accelerates the differentiation of progenitors into the astroglial lineage. Thus, various variants of NSC transformation are controlled by a certain balance of GABA/glutamatergic and signals [36].

During the development, the vertebrate telencephalon is divided into two major parts: the ventral region, or subpallium; and the dorsal region, or pallium [4]. GABA-ergic neurons originate from the subpallium and can later migrate to the pallium, as was previously reported for the zebrafish *Danio rerio* [37,38]. Thus, GABA-ergic neurotransmission dominates the nuclei of the ventral subpallial telencephalon, while glutamatergic neurotransmission dominates the pallial telencephalic region [39]. This general pattern was confirmed in a recent study on *A. burtoni* [40]. The transformation of progenitors into neurons and astrocytes is controlled by the involvement of glutamate and/or GABA specific signals expressing subtypes of the corresponding receptors [36]. In particular, the glutamatergic signal leads to inhibition of the proliferation of undifferentiated cells, while the activation of ionotropic glutamate NMDA receptors, *vice versa*, promotes the differentiation of these progenitors [35]. The activation of metabotropic type III glutamate receptors mGluR that potentiates the corresponding differentiation of astroglial cells is also a significant finding [36]. Thus, the physiologically necessary relationship between the activity of the glutamatergic and GABA-ergic systems is maintained by the balance of chemical signaling, which ensures the formation of new neurons and astrocytes of the corresponding mediator specificity [41,42].

In addition to the GS-immunopositive microcytosculpture of fibers and their terminals, aggregations of intensely labeled GS cells were also found in deep parenchymal regions at the boundaries between the pallial Dd/Dm and Dd/Dl regions (Figure 1F, inset) and areas containing an increased density of distribution of GS-immunopositive terminals in trout (Figure 1G, inset). Such areas usually showed a reduced distribution of GS+ RG fibers and/or a change in the direction of RG fibers. The maximum amount of RG in trout was found in the medial zone of the pallium and somewhat lower, in the lateral and dorsal zones (Figure 1N). The data obtained for trout differ from the results for *A. burtoni*, according to which the maximum RG distribution density is located in the lateral pallium, an area which increases in size to a greatest extent during the constitutive brain growth [26].

Thus, in the pallial zone of the trout telencephalon, we identified single GS+ cells outside the PVZ, and this finding is also consistent with the results of previous studies [25,27]. The fact that we found GS+ cells in deep regions of the pallium confirms the data for *A. burtoni* [26] and suggests that these regions receive or retain cells with stem-cell characteristics and glial function similar to superficial RG. One of the explanations may be as follows: the central niche of stem cells continues producing cells in the central pallial regions in trout. Although we cannot completely rule out this, the single cells with proliferation markers, sometimes found in this and other studies [5,6], do not support this opinion. In studies on *A. burtoni*, Sox2/GS-positive astroglial cells were found far from the ventricular surface: in the *sulcus epsiloniformis* region separating the Dc region from Dlp [43], where the ventricular surface showed a high level of proliferation [6,26]. Judging by their morphology and location, we assume these to be migrating cells.

In the trout subpallium, a dense GS+ innervation of the periventricular region was revealed in the Vd (Figure 1I,J, inset). In the lateral subpallial zone of the Vl and at the Dl/Vl border, numerous GS+ RG fibers were found, with immunonegative cells located along them (Figure 1K,L, insets). These data indicate that the central and lateral regions of the subpallium receive or retain cells with stem characteristics and glial function similar to superficial RG. Thus, in the adult trout telencephalon, stem-cell niches that maintain the production of new cells in these zones have been found in the deep layers of the pallium and subpallium.

#### 3.1.2. Expression of Doublecortin and Cell Migration in the Telencephalon

The study of immunolocalization of DC and nestin supports this point of view, as intensely labeled DC and Nes+ cells were found in the central zone of the trout subpallium (Figure 3A, inset). The most typical pattern of immunopositive cells in the trout pallium suggested the presence of DC+ cell aggregations in the PVZ and SVZ (Figure 3B–D, insets). Such a distribution of cells producing a neurodifferentiation marker is also characteristic of zebrafish [25,44] and *A. burtoni* [26]. In contrast to other mitotic markers, the doublecortin expression persists for a long time in young terminally differentiated neurons [45]. The doublecortin expression plays a decisive role in the process of axonal growth and synaptogenesis in adult animals [46]. The maximum number of DC+ cells was found in the medial subpallium of trout (Figure 3Q). Intensely labeled DC cells were detected in the central subpallium (Figure 3N, inset), as well as in the pallium (Figure 3E,F, inset). This is consistent with the results of GS immunolabeling in the subpallium, indicating that, in the deep layers of the trout pallium and subpallium, there are areas of increased RG concentration and aggregations of differentiating neuroblasts located in parenchymal areas outside the matrix zones of the telencephalon. DC+ cells have been found in the telencephalon of adult *Nothobranchius furzeri* [47] and *Astatotilapia burtoni* [26]. A high concentration of DC+ cells is characteristic of the pallial regions of the vertebrate brain; the caudal regions of the pallium, as a rule, are labeled more intensely than the rostral regions [45]. An estimation of the ratio of GS+ RG and DC+ neuroblasts in different zones of the trout pallium showed a significant prevalence of neuroblasts over GS+ RG (Figure 3R). An estimation of the ratio of GS+ cells without radial processes located in the pallial parenchyma showed that the number of such cells in the medial and central zones is significantly smaller than that of differentiating neuroblasts (Figure 3S). At last, a comparison of the total numbers of GS+ and DC+ cells in the trout pallium showed even smaller differences between the Dm and Dc (Figure 3T).

In most of the telencephalic proliferative zones in zebrafish, newborn cells diverge radially into the surrounding brain tissue and differentiate into neurons expressing HuCD and position-specific mature neuronal markers, e.g., Pax6a in the subpallium and parvalbumin in the pallium [4]. It has also been shown that neonatal neurons form SV2 synapses, thus indicating the functional integration of these cells into the brain tissue [48,49]. Since the distribution zones create different types of neuronal cells, it is important to know the molecular identity of the progenitor/stem cells populations. In rodents, reptiles, and birds, mutations in progenitor/stem cells populations demonstrate distinct glial phenotypes of radial glia or astrocytes, most of which are in contact with the ventricular lumen [4,44]. In zebrafish, the two telencephalic proliferation zones differ in cell type and marker-gene expression. In zebrafish pallium, a heterogeneous pattern of distribution of dividing and non-dividing cells was revealed on the surface of the dorsal ventricle of the telencephalon [4,48]. About two-thirds of proliferating progenitors in zebrafish exhibit a high expression of glial markers; however, various combinations of canonical glia and radial glia markers, such as GFAP, Vim, S100β, and Aro-B, have also been found [48,49]. One-third of the proliferating cells are non-glial.

Cells with positive glial immunolabeling in zebrafish exhibit a radial phenotype with long radial processes, whose ends reach the surface of *pia mater*, close to the blood vessels on the lateral surface of the dorsal telencephalon. In contrast, GS+ and Vim+ cells in the trout telencephalon often exhibit both the RG and NE phenotypes, which differ from those in zebrafish. Cells with negative glial labeling in zebrafish may be neuroblasts in a state of differentiation [25,50,51], since these characteristics are similar to the glia-negative neuroblasts proliferating in the SVZ in mice [52]. Alternatively, such cells in the trout telencephalon may represent a different cell type, e.g., immunospecific cells or progenitor cells.

#### 3.1.3. Expression of Vimentin and Glial Cell Specialization in the Telencephalon

The vimentin immunolocalization patterns in the trout telencephalon showed localization predominantly in PVZ glial cells without processes and also extracellular granular localization (Figure 5B,C, insets). Vimentin is a molecular marker of intermediate filaments in fish brain cells that is often found in immature astrocytes and their differentiated forms [53,54]. In trout, Vim+ RG cells were detected in limited areas of the dorsal and lateral pallium (Figure 5A,D, inset), and RG processes were located at a very small distance from the PVZ. The ratio of Vim+ glial cells without processes and RG in these areas showed a significant predominance of cells without processes (Figure 5J). Vim+ glial cells were not found in the central areas of the trout pallium and subpallium. A comparative analysis of the ratio of GS+ and Vim+ RG in the pallium showed a predominance of GS+ RG (Figure 5K), while an analysis of the ratio of GS+ and Vim+ glial cells without processes revealed a predominance of GS+ cells without processes only in the dorsal pallium (Figure 5L). Thus, the ratio of GS+ and Vim+ glial cells in the trout pallium suggests a predominance of GS+ over Vim+ progenitors, which is consistent with the results of a study on juvenile *Oncorhynchus masou* [8] and *Chelon labrosus* [54]. Vimentin-expressing cells of the glial type have been found in the pallium and subpallium of zebrafish [53], tanycytes of the percomorphs *Trachinotus blochii* [55], and the cartilaginous fish *Scyliorhinus stellaris* [56]. However, in the pallium of intact juvenile *O. masou*, most Vim+ cells have a NE phenotype [8]. In the subpallium of early larvae of the grey mullet, *Chelon labrosus*, weakly immunopositive RG cells have been found [54]. In later larvae of *C. labrosus*, Vim-expression is more pronounced, reaching its maximum value in large larvae [54]. In the pallium of juvenile *O. masou*, the vimentin expression is relatively low [8]; in the subpallial region of intact juveniles, along with clusters of immunopositive cells in the PVZ, Vim+ cells were detected also in the SVZ and PZ (unpublished data).

The Notch-signaling transduction cascade by Tg expression (her4.1: GFP) was visualized in zebrafish [57]. Notch signaling is active in all pallial glial cells expressing S100β, regardless whether they are proliferating progenitors or exhibit a resting phenotype. Therefore, active Notch signaling may reflect the development of glial cell specialization. However, a high level of Notch signaling, in contrast to a low one, provides a choice between the resting and active states of NSC. This has been demonstrated as the choice of the fate of embryonic NE cells during the division of neurogenic cells in the neural tube of zebrafish [50]. An imbalance in the transmission of Notch signaling affects the processes of self-renewal/differentiation in daughter cells [58]. While Notch signaling in pallial RG provides a potential mechanism that can determine the fate of NSC in the pallium, the situation in the subpallium of zebrafish and other fish species is not yet clear. The subpallial growth zone is characterized by a decrease in the expression of the glial marker and a decrease in the expression of Tg (her4.1: GFP) [57]. Nevertheless, the expression of GFP-nestin in the subpallial proliferation zone has been reliably found in transgenic zebrafish [59]. The nestin expression of proliferating precursors in the subpallial proliferative zone is shown to reflect the traits of pseudo-stratified NE cells, in particular, interkinetic migration of the nucleus, the expression of the apical marker (zona occludens 1, ZO1), and the contact of their basal processes with the pial surface [51]. Thus, different populations of pallium and subpallium progenitor/stem cells give rise to different types of neurons and express different molecular markers.

#### 3.1.4. Expression of Nestin in Telencephalon Progenitor Cells and Radial Glia

Nestin is a molecular marker of progenitor cells and RG that exhibit self-renewal and multipotency [25,60,61,62]. Teleost fish have many progenitor cells and, in contrast to mammals, have a more pronounced ability to regenerate neurons after injury [63]. Nestin-positive cells are capable of producing neurons and glial cells after cell differentiation [61,64]. In adult mammals, the subependymal zone of the lateral ventricle and the subgranular zone of *dentate gyrus* in the hippocampus are the only proliferative regions of the brain that give rise to new neurons [3,4]. On the other hand, teleost fish have many neurogenic regions, especially in the telencephalon, which contain many progenitors and stem cells. Various studies have shown that the telencephalon has a remarkable neurogenetic ability [25,61]. Proliferative regions are identified as the ventricular zones of the telencephalon and diencephalon, the border between the midbrain and hindbrain, and the ciliary marginal zone of the retina in teleosts [60].

The study of the localization of nestin in the trout telencephalon showed more extensive patterns of distribution in the PVZ (Figure 7A–D). However, intensely labeled Nes+ cells were found in deep zones of the trout pallium: Dm (Figure 7C, inset), Dc (Figure 7E,F, inset), Dl (Figure 7H, inset), and Dlp (Figure 7I, inset). In all of these zones of the pallium, as well as in the Vd and Vv of the subpallium, Nes+ progenitors were identified that are involved in the constitutive neurogenesis of the trout telencephalon outside the matrix PVZ. The maximum number of Nes+ cells was found in the medial zone of the pallium (Figure 7M), as well as in the Vd and Vc zones of the subpallium (Figure 7M), thus indicating the presence of a large number of progenitors in these regions of the trout telencephalon. A comparative analysis of distribution of GS, DC, and Nes in the pallial areas showed the dominance of GS+ cells in the Dm and Dc areas (Figure 7N), while an estimation of the ratio of Nes+ and DC+ cells revealed a significant dominance of DC+ cells in the Dc pallial and Vv subpallial areas (Figure 7O). Thus, the distribution of Nes+ cells in the trout telencephalon includes a large population of immunopositive progenitors, which probably suggests the presence of several discrete zones producing new cells during the constitutive growth in trout.

The ultrastructure of neurogenic brain niches, which are a non-ependymal layer separating the SVZ from the ventricular lumen, as in mammals, has recently been characterized in zebrafish [65]. In addition, among various types of proliferating cells that can be distinguished in the ventricular zone, none bear cilia protruding into the ventricular lumen. Furthermore, pallial and subpallial progenitor/stem cells are retained within the differential expression region of transcription factors [63], thus also reflecting a diversity of progenitors: RG *vs*. NE cells. However, in terms of development, RG originate from NE cells, whose basal processes have grown longer due to the tissue growth during the brain development [11]; both are true epithelial cells. A variety of progenitors and a limited progenitor potential have been observed in rodents, whose progenitors that give rise to different types of neurons show differential expression of glial markers [11,66]. Thus, the emergence of different types of progenitors with a pronounced potential may be more widespread than the emergence of a single NSC population in zebrafish and trout.

### 3.2. Tectum Opticum

#### 3.2.1. Expression of Glutamine Synthetase in the Tectum

In the trout tectum, GS+ cells were found in the marginal and periventricular layers (Figure 2A, inset). Among GS+ cells, we identified both RG cells of two types and immunopositive cells without processes, representing cells of the NE type. The arrangement of these cells corresponded to the matrix zones of the tectum, the periventricular (PGZ) and the outer marginal zone (MZ). In trout, RG cells often formed clusters separated by immunonegative regions in the marginal zone (Figure 2B,D). The presence of GS+ RG cells was revealed also in other fish species such as *Apteronotus albifrons* [34], the zebrafish [67], and the chum salmon *O. keta* [23]. Tectal glia have been described as “tanycytes” from the ocean sunfish, *Mola mola* [68]; however, cells of this type in the goldfish, *Carassius auratus*, were referred to as “radial glia” [69], and in *L. macrochirus* and *P. nebulifer* as “ependymoglia” [70].

The tectal growth zone around the posterior half of the optic nerve, located at the edge of the peripheral germinal zone (PGZ) facing the tectal ventricle, has been described from the zebrafish [44,71], goldfish, and the medaka, *Oryzias latipes* [72,73]. Experiments with S-phase markers have shown that cells formed in this proliferation zone are added concentrically to the PGZ, thereby causing the tectum to grow [71]. Cell types produced by the tectal proliferative zone differentiate into glutamatergic and GABA-ergic neurons, as well as other cell types: HuCD+ expressing neurons, oligodendrocytes, and RG. 

In the trout *O. mykiss*, an increased density of GS+ fibers was revealed in the *stratum griseum centrale* (SGC) and the *stratum griseum et album periventriculare* (SGAP); varicose thickenings extended along the fibers, forming areas of increased glutamatergic innervation (Figure 2B, in white rectangle). We associate the presence of GS+ innervation with the signaling suppressing the proliferation of undifferentiated neural progenitors [36]. On the other hand, the presence of varicose glutamatergic fibers in the trout tectum may be associated with the differentiation of precursors into glial cells. This is confirmed by the results of DC labeling in trout (Figure 2C,E), which revealed the presence of large neurogenic aggregations in the tectum parenchyma. RG cells become integrated into the ventricular ependymoglia sheet, which separates the PGZ from the ventricle, and are located in the periventricular zone of the telencephalon. These ependymoglial cells are directed toward the pial surface and express canonical RG markers, such as GFAP, S100β, and BLBP [21]. Studies on different fish species have shown that these cells are not widely distributed [4]. It has been found that they resume proliferation after damage to the optic nerve in trout [74]. On the other hand, the cells of the tectum marginal zone express progenitor markers, such as Sox2 or Musashi1, and do not exhibit the properties of RG in zebrafish and medaka [22]. Instead, they discretely express polar markers, namely ZO1, gamma-tubulin, and aPKA, thus suggesting that these progenitors/stem cells are of the NE type [4].

In trout, GS+ cells of MZ contain both NE cells and RG cells (Figure 2B,D). The number of RG cells in the MZ of the lateral and dorsal tectum is greater than that in the PGZ, but no significant differences were found (Figure 2J). Nevertheless, the ratio of the numbers of GS+ cells of the NE type in the MS of the dorsal tectum significantly (*p* < 0.05) exceeded that in the lateral tectum (Figure 2K). The number of NE-type GS+ cells in PGZ was multifold greater than that in SM; in the lateral tectum, the concentration of NE-type GS+ cells was significantly higher (*p* < 0.01) than that in the dorsal one (Figure 2K). In the trout PGZ, the dominant type of GS+ cells is NE, with their number in the lateral tectum being significantly greater (*p* < 0.05) than in the dorsal one (Figure 2L). In the MS, the ratio of GS+ RG and NE cells in the dorsal tectum is approximately the same, but GS+ RG dominates the lateral tectum (Figure 2M). The total number of GS+ cells of both types in the dorsal tectum is significantly greater (*p* < 0.05) than that in the lateral (Figure 2M). Thus, in contrast to zebrafish and medaka, a large number of RG cells were found in the MZ in trout; meanwhile, in the PGZ, GS+ cells of the NE type dominated.

It is unknown whether they constitute differential molecular sets of these precursors that are assumed to express various other transcriptional regulators. Further analysis should clarify whether this population of cells contains pluripotent stem-cell sets or differentially defined progenitors that are grouped together. Regardless, it is of interest to study the mechanisms that cause the emergence of precursors of the NE type in the marginal PGZ and the RG type along the intertectal ventricle.

#### 3.2.2. Expression of Doublecortin in the Tectum

DC expression was detected in various types of trout tectum cells, thus indicating the occurrence of such cells in the adult period of development and the incorporation of these cells into existing neuronal networks. The maximum number of DC+ cells was found in the area of the PGZ, where such cells were located among precursors of the NE type (Figure 4B,D). The results of a comparative analysis of the distribution of DC+ cells in the dorsal and lateral tectum confirm this finding (Figure 4J). Intensely labeled DC cells were identified along the RG fibers, apparently representing a migrating population of differentiating neuroblasts directed into the deep layers of the trout tectum (Figure 4C, inset). A study on *Nothobranchius furzeri* showed that it takes 7 days to embed new neurons into the tectum in this species [47]. In *D. rerio*, almost 50% of the total number of proliferating cells differentiate into neurons on days 270–744 [5,75]. In the electric fish, *Gymnotus omarorum*, a shorter range or fewer migratory processes involved in multimodal processing of visual information have been found in the optic tectum and optic tract. The migration process of newborn cells in the tectum of *G. omarorum* occurs mainly from the caudal pole of the tectal zone of proliferation toward almost all layers of the tectum, as well as from the dorsomedial edge of the more rostrally located zones of the tectal zone of proliferation [13]. This process includes the addition of cells to the dorsomedial edge of the tectal proliferation zone and the displacement of older cells to the adjacent caudal parts of TeO in the ventrolateral direction, as is consistent with the data for *Oryzias latipes* [22], *C. auratus* [72], *D. rerio* [71], and *N. furzeri* [47]. The network of DC+ processes is located under the dorsomedial tectal proliferative zone in *G. omarorum* [13].

#### 3.2.3. Expression of Vimentin and Neuroregenerative Properties of Tectum

In the fish brain, glial cells are morphologically and immunohistochemically different from mammalian astrocytes [76]. This difference confuses the nomenclature for the same glial cell type in fish and mammals. Fish glial cells, although commonly referred to as “astroglial”, have fewer intermediate filamentous bundles than mammalian astroglial cells and, in some cases, even lack such bundles. Because of their radial orientation, these cells are termed radial glia [77] or radial astrocytes [78]. Radial astrocytes are considered to be the only non-ependymal astroglial cells in the adult spinal cord [79]. To avoid confusion, Cuoghi and Mola [76] suggest using the term “astrocyte-like cells” and generally consider immunohistochemical characteristics as identification criteria.

In the trout tectum, the number of Vim+ cells is rather limited, and the level of vimentin immunopositivity is relatively low (Figure 6A). However, several types of Vim+ cells of the NE type were identified in SM; above SM, neurogenic niches of various sizes were found in the ependyma (Figure 6B, inset). Vim+ ependymoglial-type cells without processes were found in the SM and PVZ, where they were included in various constitutive neurogenic groups (Figure 6F, inset; and Figure 6G,H).

A comparison of the numbers of astrocyte-like cells expressing Vim in the dorsal and lateral tectum of trout showed the number of Vim+ in PVZ to be significantly greater (*p* < 0.05) than that in SM (Figure 6J). A comparative analysis of the distribution of Vim and GS in the lateral tectum showed a significant dominance of Vim+ cells (*p* < 0.05); in the dorsal tectum, a dominance (*p* < 0.05) of GS+ cells was observed (Figure 6K). An estimation of the ratio of Vim and GS expression revealed a significant predominance of the lateral tectum (*p* < 0.001) cells over the dorsal tectum cells in the PVZ in trout (Figure 6L). We associate this fact with the presence of the laterocaudal matrix zone of the tectum, which is involved in constitutive neurogenesis. Vimentin-producing tanycytes have been identified in several brain regions of juvenile and adult mullet, *Chelon labrosus* [54]. The function of tanycytes is unclear; however, being associated with the ventricles and blood vessels, they are assumed to play a role in the perception of the biochemical composition of both cerebral fluid and blood and transmit this information to neural elements [68]. Since tanycytes in *M. mola* [68] and *O. masou* contain NADPH-diaphorase [80], which is a marker of nitric oxide synthase, they are suggested to modulate the activity of nearby neuronal circuits. In *Barbus comiza*, radial astrocytes are mainly GFAP-immunoreactive, but single vimentin-immunopositive ependymocytes can occasionally be observed in the ventral region of the spinal cord. Processes of these cells terminate in the subpial zone, forming a continuous subpial glia [79].

A study of the differential expression of transcription factors (TFs) after injury to the tectum in medaka and zebrafish showed that the expression of some pro-regenerative TFs is induced only in the zebrafish tectum [21]. In addition, it was found that the expression of *sox2*, *stat3*, and *oct4* is required for the proliferation and differentiation of NSCs into neurons [81,82]. The expression of these TFs has also been investigated in order to assess the potential changes induced in response to injury to the tectum. It was shown that the TFs *ascl1a* and *oct4* are induced after a damage to the medaka tectum [21].

The regenerative abilities of the zebrafish tectum were found to significantly exceed those of the medaka tectum, despite the similarities in the structure of the brain, body size, and life expectancy [30]. Studies of traumatic injury to the tectum have shown that a stab wound can cause proliferation of RG in medaka, but with a limited generation of new neurons at the site of injury compared to the response observed in zebrafish [21]. Thus, comparative studies have shown unequal capacities for neuronal regeneration in the adult brain between various teleost fish species.

An investigation of the distribution pattern of GFAP+ RG fibers appearing at 2 weeks after an injury to the tectum showed that a glial scar-like structure with the absence of a cell layer covers the injured area in medaka. These data indicate that medaka has a low regenerative capacity in the tectum compared to zebrafish, because RG in the injured medaka tectum may have reactive astrocytic characteristics rather than neurogenic NSCs [21]. In addition to the limited generation of neurons after the tectum injury in medaka, persistent GFAP+ and BLBP+ scar-like structures were recorded between days 14 and 30 post-injury. In adult zebrafish, a stab wound to the telencephalon caused reactive gliosis with increased regulation of GFAP immunoreactivity, but no scarring was observed [28,83]. In injured zebrafish tectum, despite the recorded increased GFAP immunoreactivity, no overt scarring similar to that in medaka was recorded. This suggests that the scar structures with radial fibers in the damaged tectum in medaka are similar to the glial scar formed by reactive astrocytes in the damaged mammalian CNS [84]. These results show that the differential expression of pro-regenerative factors may contribute to limiting the RG neuron-differentiation potential in some fish species—in particular, medaka—during tectum regeneration.

#### 3.2.4. Expression of Nestin in the Tectum

Nestin is a type VI intermediate filament protein found primarily in postmitotic cells [60,61,64]. The function of nestin and the nestin activation observed after injury may be a key to understanding cell regeneration. These data were obtained through studies of nestin expression in goldfish, a species with an exceptional capability of neurogenesis in adults and regeneration of the CNS [85]. The research results have shown a different role of nestin in the processes of neurogenesis and neuronal regeneration [63]. Several studies have reported that nestin can affect the ability of a cell to move particles (e.g., vesicles) intracellularly, thereby providing structural and functional support during cell proliferation [60,62,86]. Nestin is involved in the control of localization and separation of intermediate fibers within the cytoskeleton, thereby influencing the distribution of cellular components during mitosis [60,86]. Nestin plays an important role in the reorganization of cytoskeletal filaments, thus controlling the cellular dynamics by regulating the polymerization of various intermediate filaments [61]. As has been found by using Western blotting, nestin immunoreactivity is present in all the brain regions of adult goldfish, similarly to the situation observed during the mammalian CNS development [63,87]. In contrast, the distribution of nestin in the adult mammalian CNS is limited to only a few proliferative regions [60,61,64,86].

In the trout tectum, a significant level of nestin immunopositivity was revealed in various types of cells and layers (Figure 8A–C, insets). The concentration of Nes+ cells, as high as in the matrix areas of the tectum in the PVZ and SM, and also the expression of Nes in the RG in the trout tectum indicated a high intensity of constitutive neurogenesis. As with DC immunolabeling, the nestin labeling revealed extensive areas of parenchymal neurogenesis to which congregations of Nes+ cells usually adjoined (Figure 8B, inset). The extensive distribution of Nes+ cells in the composition of differentiated layers of SGC (Figure 8D), as well as the presence of patterns of intracellular and extracellular granular expression of nestin, indicated a wide involvement of this protein in the constitutive neurogenesis of the trout tectum.

A quantitative assessment of the proportions of the numbers of GS+, Vim+, and Nes+ NE type cells in the lateral and dorsal tectum of trout revealed an interesting relationship between cells in the PVZ and SM (Figure 8K). The ANOVA showed significant intergroup differences (*p* < 0.01) in the distribution of GS+ and Nes+ cells and significant intergroup differences (*p* < 0.05) in the distribution of Vim+ and Nes+ cells in the SGP of the lateral tectum, indicating the dominance of GS+ ependymoglial progenitors. In the dorsal tectum, on the contrary, significant intergroup differences (*p* < 0.01) were found in the distribution of Nes+ and GS+ cells and significant intergroup differences (*p* < 0.05) in the distribution of Vim+ and GS+ ependymoglial cells. The data obtained indicate the predominance of ependymoglial progenitors over Nes+ cells in the trout dorsal tectum.

An estimation of the ratio of the numbers of Nes+ and GS+ precursors of the NE and RG types in the PVZ and SM of the lateral and dorsal tectum showed significant intergroup differences in the distribution of precursors of various cell phenotypes (Figure 8L). In the MS of the dorsal tectum, the number of Nes+ precursors of the ependymoglial type was significantly greater than that of GS+ (*p* < 0.05); for precursors of the RG type, on the contrary, the number of GS + cells was greater (*p* < 0.05) than that of Nes+ progenitors (Figure 8L). In the SGP of the lateral tectum, the number of Nes+ ependymoglial progenitors was significantly greater (*p* < 0.05) than the number of GS+ ependymoglial progenitors (Figure 8L).

In adult zebrafish, NSCs in the tectum are characterized by certain features, such as, in particular, silent RG under normal physiological conditions; NSCs are activated as a result of trauma, with their subsequent proliferation and differentiation [88]. The zebrafish tectum also contains NE-type cells that continuously proliferate and generate new neurons throughout life [71,89]. Thus, the high number of Nes+ cells in the trout tectum, as in the region of the matrix areas of the tectum in the PVZ and SM, and the expression of Nes in the RG in the trout tectum are consistent with the data for zebrafish and indicate a high intensity of constitutive neurogenesis.

Earlier, we studied the nestin expression in the cerebellum after a traumatic injury in juvenile *O. masou*. As a result of the cerebellum injury, the expression of nestin was repeatedly detected in all areas of the cerebellar body, especially in the lateral zone. Although several functional properties of nestin have been assumed, the mechanism of its action at the molecular level is still unclear. The discovery of new nestin isoforms in goldfish [63] may deepen the understanding of the neurogenic potential of the vertebrate brain. Evidence exists that the nestin protein sequence contains a short amino terminus and a long carboxyl terminus and is fairly well conserved among various species, such as chicken, mouse, rat, zebrafish, frog, and human [61,86]. Zebrafish morphants, which had been injected with a nestin morpholino, exhibited severe malformations, including morphological abnormalities: body shrinkage and head growth; small eyes; underdevelopment of the retinal lens; brain defects; and defects in the development of motor neurons, axons, and glial cells in the brain [86]. Such morphants also showed reduced niches of proliferative cells in the developing nervous system that led to a decrease in the number of progenitor cells, possibly caused by cell apoptosis and death [86]. There is also evidence that nestin is involved in a mechanism controlling the neurokinetic capacity and cellular regenerative processes. Characterization of nestin in teleosts may provide a better understanding of the mechanisms underlying the brain plasticity. Thus, the study of the nestin expression in the trout brain can be considered as an additional effective approach to elucidating the constitutive neurogenesis.

## 4. Material and Methods

### 4.1. Experimental Animals

The study was carried out on 20 three-year-old individuals of the rainbow trout, *Oncorhynchus mykiss*, with a body length of 27–36 cm and a body weight of 285–320 g. The animals were obtained from the Ryazanovka experimental fish hatchery in 2020. The fish were kept in tanks (200 × 150 × 70 cm, five fish per tank) with aerated fresh running water, at a temperature 14–15 °C, and fed once a day. The daily light/dark cycle was 14/10 h. The concentration of dissolved oxygen in the water was 7–10 mg/dm^3^, which corresponded to normal saturation. All experimental manipulations with the animals were in compliance with the rules listed in the charter of the A.V. Zhirmunsky National Scientific Center of Marine Biology (NSCMB) FEB RAS and the Ethical Commission regulating the humane treatment of experimental animals (approval # 3-171221 from Meeting No. 3 of the Commission on the biomedical ethics at NSCMB FEB RAS, 17 December 2021).

### 4.2. Preparation of Material for Immunohistochemical Studies

Anesthesia and prefixation were carried out as follows. The animals that were removed from the experiment were anesthetized in a 0.1% solution of ethyl-3-aminobenzoate methanesulfonate (MS222) (Sigma, St. Louis, MO, USA, Cat. # 1219 WXBC9102V) for 10–15 min and euthanized by the method of rapid decapitation. After the anesthesia, the intracranial cavity of the immobilized animals was perfused with a 4% paraformaldehyde solution prepared in 0.1 M phosphate buffer (pH 7.2). After the prefixation, the brain was extracted from the cranial cavity and fixed in the 4% paraformaldehyde solution for 2 h at 4 °C. Then it was kept in a 30% sucrose solution at 4 °C for two days (with the solution changed seven times). Serial frontal brain sections (50 μm) were cut on a freezing microtome (Cryo-star HM 560 MV, Oberkochen, Germany). Every third frontal section of the telencephalon and tectum was taken for the reaction. 

### 4.3. Immunohistochemistry

To study the expression of glutamine synthetase, doublecortin, nestin, and vimentin in the telencephalon and tectum of *O. mykiss*, immunoperoxidase labeling was performed on frozen free-floating brain sections. The brain sections were pre-incubated for 30 min at room temperature in PBS supplemented with 10% non-immune horse serum, 0.01% Tween 20 (Sigma, St. Louis, MO, USA), and 0.1% BSA (Sigma, St. Louis, MO, USA). Monoclonal mouse antibodies against doublecortin (catalog number CO613 sc-271390; Santa Cruz Biotechnology, CA, USA), GS (catalog number GF5 ab10062), vimentin (catalog number 3B4 ab28028), and nestin (catalog number 2C1.3A11 ab18102) (Abcam, Cambridge, UK) at a dilution of 1:300 were applied to the frontal 50 μm sections incubated in situ at 4 °C for 48 h. To identify reaction products, a red substrate (VIP Substrate Kit, Vector Laboratories, Burlingame, CA, USA) was used in accordance with the manufacturer’s recommendations. The brain sections were placed on polylysine-coated glass slides (BioVitrum, St. Petersburg, Russia) and left to dry completely. For identification of immune-negative cells, the brain sections were additionally stained with a 0.1% methyl green solution (Bioenno, Lifescience, CA, USA, Cat # 003027). The color development was monitored within 15 min under a microscope. The sections were washed in three changes of distilled water for 10 s, differentiated in a 70% alcohol solution for 1–2 min, and then in 96% ethanol for 10 s. The brain sections were dehydrated in accordance with the standard procedure: placed in two changes of xylene, 15 min per each, and then embedded in the Bio-Optica medium (Milano, Italy) under coverslips.

### 4.4. Microscopy

Cell bodies were visualized, and a morphological and morphometric analysis of their parameters (measurements of the greater and lesser diameters of the soma) was carried out under a Zeiss Axiovert 200 M fluorescence motorized phase contrast microscope with an ApoTome fluorescence module and AxioCam MRM and AxioCam HRC digital cameras (Carl Zeiss, Germany). The material was analyzed by using the AxioVision software. Measurements were performed at magnifications of 100×, 200×, and 400× in several randomly selected fields of view for each study area. The number of labeled cells in the field of view was counted at a magnification of 200×. Micrographs of the sections were taken with an Axiovert 200 digital camera. The material was processed by using the Axioimager program and the Corel Photo-Paint 12 graphics editor.

### 4.5. Densitometry

The optical density (OD) of IHC labeling products in neuronal bodies and immunopositive granules was measured by using the Axiovert 200-M microscope software. For this, a standard evaluation of optical density for 5–7 sections, with 10–15 intensely/moderately labeled and immunonegative cells of the same type selected for analysis, was conducted in the Wizard program. Then the average OD value for each type of cells was subtracted from the maximum OD value for immunonegative cells (background), and the actual values were expressed in terms of relative units of optical density (UOD). OD in immunopositive cells was categorized into high (180–130 UOD, designated as +++), moderate (130–80 UOD, corresponding to ++), weak (80–40 UOD, corresponding to +), and low (less than 40 UOD, corresponding to −); the initial OD value was measured on the control mounts. Levels of glutamine synthetase, nestin, vimentin, and doublecortin activity in cells were determined on the basis of data of the densitometric analysis. These data, along with the dimensional characteristics, were used for typing cells on the basis of the previously developed classification of cells in the pallial zone of the *O. mykiss* telencephalon [74] formed during the period of constitutive and reparative neurogenesis.

### 4.6. Statistical Analysis

The quantitative processing of morphometric data of IHC labeling was performed by using the Statistica 12, Microsoft Excel 2010, and STATA (StataCorp. 2012, Stata Statistical Software: Release 12. College Station, TX: StataCorp LP, USA) software packages. All data are presented as mean ± standard deviation (M ± SD) and were analyzed by using the SPSS software application (version 16.0; SPSS Inc., Chicago, IL, USA). All changes in the group were compared by using the one-way or two-way analysis of variance (ANOVA, Chicago, IL, USA) with Bonferroni’s correction. Values at *p* ≤ 0.01 and *p* ≤ 0.05 were considered statistically significant.

## 5. Conclusions

The results of the study have shown that the *telencephalon* and *tectum opticum* of adult trout contain different types of cells expressing glutamine synthetase, doublecortin, vimentin, and nestin. In general, the patterns of distribution of GS+ and DC+ structures in the telencephalon and tectum in trout resemble those in other fishes, in particular, *Astatotilapia burtoni*. However, some additional features not characteristic of other studied fish species have been revealed in trout, including the terminal apparatus intensely immunolabeled with GS that braids surfaces of immunonegative cells and represents the terminal targets of glutamatergic innervation. We associate the presence of GS+ innervation with the signaling suppressing the proliferation of undifferentiated neural progenitors [36]. On the other hand, the presence of varicose glutamatergic fibers in the trout tectum may be associated with the differentiation of precursors into glial cells. This is confirmed by the results of DC labeling in trout that revealed large neurogenic aggregations in the tectal parenchyma. RG cells integrate into the ventricular ependymoglia sheet, which separates the PVZ from the ventricle, and are located in the periventricular zone of the telencephalon. These ependymoglial cells are directed toward the pial surface and express canonical RG markers, such as GFAP, S100β, and BLBP [21]. The results of studies on different fish species have shown that such cells are not widely distributed [4]. It is still unknown whether they constitute differential molecular precursor sets that are assumed to express other various transcriptional regulators. Further analysis should clarify whether this cell population contains pluripotent stem-cell sets or differentially defined progenitors that are grouped together. Anyway, the study of the mechanisms responsible for the emergence of NE type progenitors in the marginal zone and the RG type along the dorsal telencephalic and tectal ventricles is of considerable interest.

The vimentin expression in the trout telencephalon and tectum is limited to two periventricular surface zones and is almost not represented in the RG, which is significantly different from the expression in other fish species. The most common type of Vim+ cells in the telencephalon and tectum is the NE type. On the contrary, the expression of nestin in the telencephalon and tectum is characterized by a high variety of cell forms of not only superficial, but also parenchymal, localization. The different types of Nes+ precursors of the NE and RG types, whether single or forming zones of constitutive neurogenesis, indicate a high neurogenic potential in the telencephalon and tectum of rainbow trout, *O. mykiss*.

## Figures and Tables

**Figure 1 ijms-23-01188-f001:**
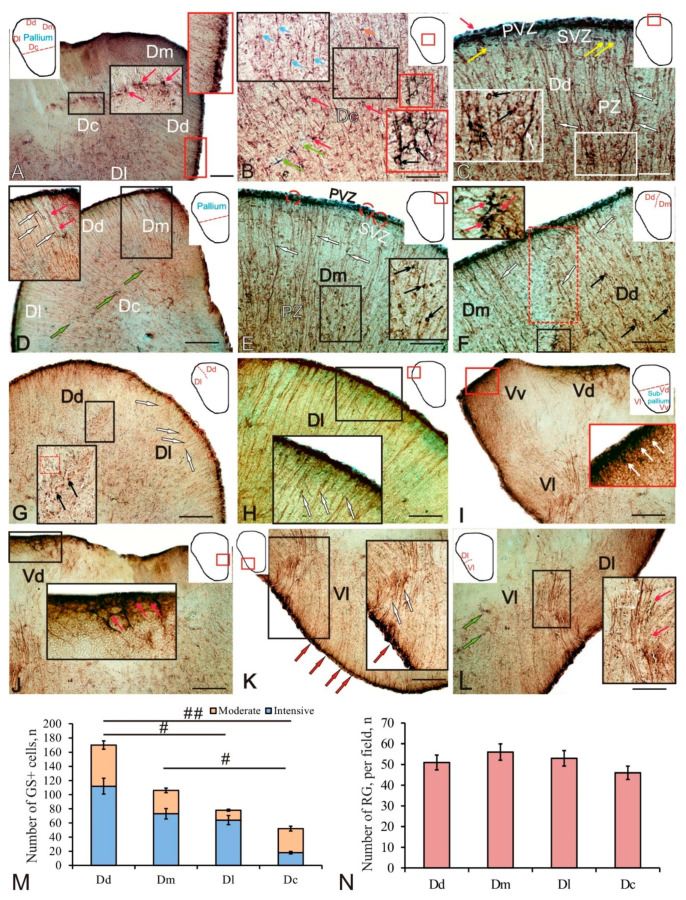
Glutamine synthetase (GS) in the pallial and subpallial regions of the telencephalon in trout, *Oncorhynchus mykiss*: (**A**) A general view of GS immunolocalization patterns in the trout pallium (Dd—dorsal, Dm—medial, Dl—lateral, and Dc—central zones of pallium); the pictogram shows the zones of the dorsal telencephalon (pallium), the dotted line separates the pallium from the subpallium, aggregations of GS+ cells, intensely labeled cells (red arrows) in Dc (inset in black rectangle), and periventricular area zone (Dd, inset in red rectangle). (**B**) Central pallial zone (Dc) at a higher magnification; inset (in black rectangle) shows a fragment including radial glia (light blue arrows), moderately labeled GS+ cells (red arrows), vessels (green arrows), and inset in red rectangle showing an intensely labeled terminal apparatus of glutamatergic fibers (black arrows) covering GS+ cells. (**C**) Dorsal pallial zone (Dd) intensely labeled GS+ cells of the NE type (red arrow) in the periventricular zone (PVZ), moderately labeled cells (yellow arrows) in the subventricular zone (SVZ), GS+ radial glia (white arrows) in the parenchymal zone (PZ), and heterogeneous cell complexes (inset), including immunonegative cells adjoining clusters of small moderately labeled GS+ cells (black arrows) and/or nerve terminals. (**D**) Distribution of GS+ radial glia (white arrows) in the pallium, forming guidelines for migration (pink arrows) of mature cells (inset), and vessels (green arrows). (**E**) Intensely labeled groups of NE cells (in red dotted oval) in the medial (Dm) zone of the pallium in the PVZ and intensely labeled terminals (black arrows) of glutamatergic fibers (inset) in the PZ. (**F**) A change in the direction and density of RG distribution (white arrows) is revealed at the border between Dm and Dd (in red dotted rectangle), and an area of increased distribution density of intensely labeled GS+ terminals (pink arrows) is located ventrally (inset). (**G**) A border area between Dl and Dd (inset) and a cluster pattern of distribution of GS+ cells (in red dotted oval), GS+ fibers (white arrows), and their terminals (black arrows) on GS-negative (GS–) cells (in red dotted rectangle) revealed in Dl. (**H**) GS+ RG cells (inset, white arrows) in Dl. (**I**) General view of GS immunolocalization patterns in subpallium (Vd—dorsal, Vv—ventral, and Vl—lateral zones of subpallium); the pictogram shows the zones of the ventral telencephalon (subpallium) and intensive GS labeling of NE cells in Vv (inset). (**J**) GS+ cells in the periventricular zone of the VD (inset) and an intensely labeled neuropil detected in the SVZ (pink arrows). (**K**) GS+ RG fibers (white arrows) in Vl (inset) and NE cells (red arrows) in PVZ. (**L**) RG bundles at the Dl and Vl border (inset), with immunonegative cells localized along them (pink arrows), and vessels (green arrows). Scale bars: (**A**,**D**,**G**,**I**–**L**) 200 µm and (**B**,**C**,**E**,**F**,**H**) 100 µm. (**M**) Ratio of GS+ intensely/moderately labeled immunonegative cells, significant intergroup differences, # (*p* < 0.05), ## (*p* < 0.01), between groups of intensely and moderately labeled cells in different areas of the pallium (*n* = 5 in each group), one-way ANOVA. (**N**) Number of RG cells in different zones of the pallium (M ± SD), where M is the mean and SD is the standard deviation (*n* = 5 in each group).

**Figure 2 ijms-23-01188-f002:**
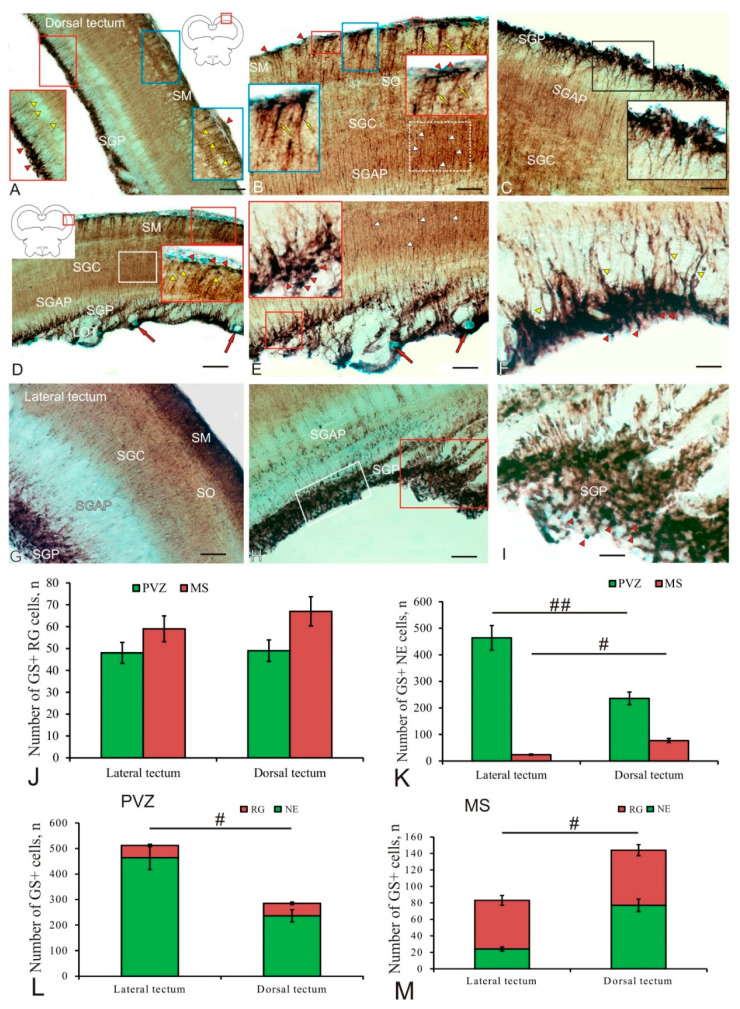
Glutamine synthetase (GS) in the dorsal and lateral *tectum opticum* of trout, *Oncorhynchus mykiss*: (**A**) A general view of GS immunolabeling in the dorsal tectum; the pictogram in the red square shows the selected tectum area, GS+ neuroepithelial (NE) progenitors (red arrowheads) in the marginal layer (SM), and RG (yellow arrowheads) identified in the periventricular gray layer (SGP) (red inset). (**B**) Dorsal part of the tectum at a higher magnification; NE cells located singly (red arrowheads) or forming small clusters (red inset) in SM, clusters of GS+ RG cells (yellow arrows) with processes extending into the inner layers of the tectum (blue inset), and an increased density of GS+ fibers (white arrowheads) with varicose thickenings (in white dashed rectangle), the layer of optical fibers (SO) revealed in the central gray layer (SGC) and the central gray and white layer (SGAP). (**C**) Periventricular aggregations of GS+ cells (inset) in the dorsal tectum (SGP—central gray layer). (**D**) GS immunolocalization in the lateral tectum, GS+ innervation region in SGC and SGAP (in white rectangle), GS+ NE cells and RG forming small clusters (red inset) in SM (designations as for (**A**,**B**)), lateral optical tract (LOT), and large vessels of the intertectal vascular plexus (red arrows). (**E**) Latero-caudal constitutive neurogenic zone of the tectum at a higher magnification and an aggregation of intensely labeled GS+ NE and RG cells (red arrowheads) in inset (other designations, as in (**D**)). (**F**) Periventricular GS+ cell populations forming constitutive neurogenic niches in the lateral zone of the tectum, containing NE cells (red arrowheads) and RG (yellow arrowheads). (**G**) General view of GS distribution in the lateral tectum. (**H**) Heterogeneous cellular composition of the lateral periventricular zone, with homogeneous areas containing NE cells (in white rectangle), and a hypertrophied zone with heterogeneous GS+ cell clusters (in red rectangle). (**I**) A section of the hypertrophied zone at a higher magnification with single NE cells (arrowheads). Scale bars: (**A**,**D**) 200 µm, (**B**,**C**,**E**,**G**,**H**) 100 µm, and (**F**,**I**) 50 µm. (**J**)—A comparative distribution of GS+ RG in the lateral and dorsal tectum (M ± SD), where M is the mean and SD is the standard deviation (*n* = 5 in each group). (**K**) A comparative distribution of GS+ NE cells in the lateral and dorsal tectum with significant intergroup differences in the number of MS cells in the lateral and dorsal tectum, # (*p* < 0.05), and the number of cells in PVZ in the lateral and dorsal tectum, ## (*p* < 0.01) (*n* = 5 in each group), one-way ANOVA. (**L**) A comparative distribution of the total number of NE and RG GS+ cells in the PVZ of the lateral and dorsal tectum, # (*p* < 0.05) (*n* = 5 in each group), one-way ANOVA. (**M**) A comparative distribution of the total number of GS+ NE and RG cells in the MZ of the lateral and dorsal tectum, # (*p* < 0.05) (*n* = 5 in each group), one-way ANOVA.

**Figure 3 ijms-23-01188-f003:**
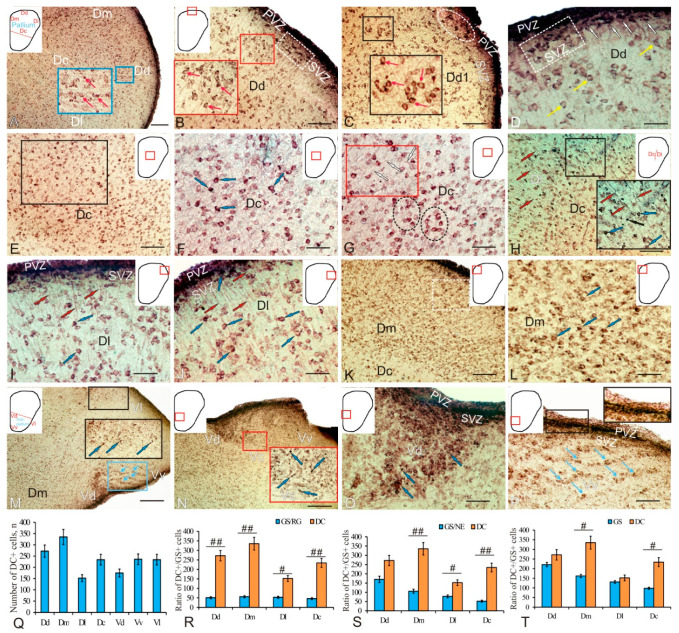
Doublecortin (DC) in the pallial and subpallial regions in the telencephalon of trout, *Oncorhynchus mykiss*: (**A**) a general view of DC immunolocalization patterns in the trout pallium (Dd—dorsal, Dm—medial, Dl—lateral, and Dc—central zones of pallium), the pictogram showing the zones of the dorsal telencephalon (pallium), with the dotted line separating the pallium from the subpallium, aggregations of DC+ cells, and intensely labeled cells (red arrows) in Dd (inset in blue rectangle); (**B**) a Dd fragment at a higher magnification, dense conglomerates of cells (in white dotted rectangle) with granular DC labeling detected in the SVZ, and diffuse clusters (red inset) of uniformly labeled cells (pink arrows) revealed in the PZ; (**C**) a fragment of Dd1 containing moderately labeled DC cells (inset) and intensely labeled cells located in the apical part of the SVZ (white dashed oval); (**D**) various types of DC+ cells of the dorsal pallium, undifferentiated intensely labeled cells (white arrows), a cluster with intensely and moderately labeled cells in the SVZ (in white dotted rectangle), and later stages of neuronal differentiation of weakly DC labeled oval and elongated cells (yellow arrows); (**E**) DC immunolocalization in the central zone of Dc pallium; (**F**) an area of E at a higher magnification (in black rectangle), with intensely and moderately DC+ labeled cells (blue arrows); (**G**) diffuse (in red rectangle) and cluster (in black dotted ovals) pattern of distribution of DC+ cells in Dc with proximal areas of neuroblast processes (white arrows); (**H**) DC+ cells at the border between Dc and Dl (inset), intensely labeled cells (black arrows), moderate (blue arrows), and RG fibers (red arrows); (**I**) DC immunolocalization in the lateral zone of the pallium (Dl); (**J**) the cluster pattern of distribution of DC+ cells in Dl (designation as in (**H**)); (**K**) DC immunolocalization in the medial (Dm) zone of the pallium at the border with Dc; (**L**) a fragment of (**K**) at a higher magnification (in white dotted rectangle) with DC+ cells (blue arrows); (**M**) a general view of DC immunolocalization in the subpallium (Vd—dorsal, Vv—ventral, and Vl—lateral zones of subpallium), the pictogram showing the zones of the ventral telencephalon (subpallium), with the dotted line separating the pallium from the subpallium, aggregation of DC+ cells in Vl (black inset), and small intensely DC labeled cells (blue arrows) in Vv (blue inset); (**N**) hypertrophied area between Vd and Vv, including the central Vc zone (red inset), with small intensely labeled and larger DC+ cells (blue arrows); (**O**) high density of distribution of DC+ cells in Vd (arrows); (**P**) in Vv, intensely and moderately DC+ cells in PVZ and SVZ (inset) and a stratified pattern of distribution of DC+ cells (blue arrows) in PV. Scale bars: (**A**,**M**,**N**) 200 µm, (**B**,**C**,**E**,**H**,**K**) 100 µm, and (**D**,**F**,**G**,**I**,**J**,**L**,**O**,**P**) 50 µm. (**Q**) A comparative distribution of DC+ cells in the pallium and subpallium of trout (M ± SD), where M is the mean and SD is the standard deviation; (**R**) a comparative distribution of DC+ and GS+ RG in the pallium with significant intergroup differences in the number of cells in Dl, # (*p* < 0.05); and the number of cells in Dd, Dm, Dc, ## (*p* < 0.01) (*n* = 5 in each group), one-way ANOVA; (**S**) a comparative distribution of DC+ and GS+ NE cells in the pallium with significant intergroup differences in the number of cells in Dl, # (*p* < 0.05); and the number of cells in Dm, Dc ## (*p* < 0.01) (*n* = 5 in each group), one-way ANOVA; (**T**) a comparative distribution of DC+ and the total number of GS+ cells in the pallium, with significant intergroup differences in the number of cells in Dm and Dc, # (*p* < 0.05) (*n* = 5 in each group), one-way ANOVA.

**Figure 4 ijms-23-01188-f004:**
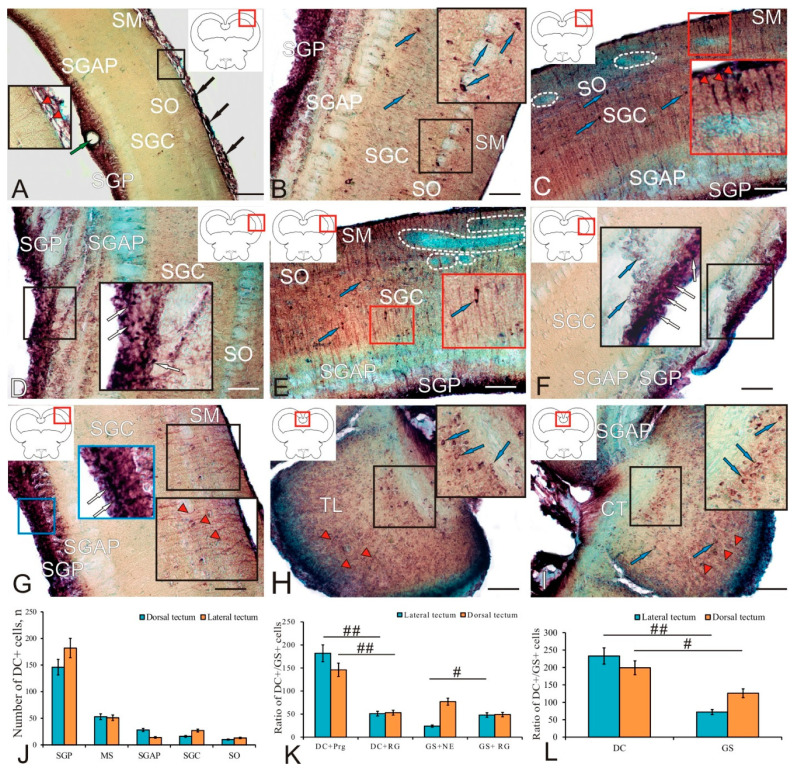
Doublecortin (DC) in the *tectum opticum* of trout, *Oncorhynchus mykiss*: (**A**) a general view of DC distribution in the dorsolateral tectum, intensely labeled cells (red arrowheads) and RG (inset) in the SM, a large vessel in the SGP (green arrow), and *pia mater* (black arrows); (**B**) DC+ cells (blue arrows) in the inner layers of the tectum (inset); (**C**) DC+ RG (red arrowheads) in SM (inset) and constitutive neurogenic niches (bounded by white dotted line) in the basal part of SM and/or SO; (**D**) patterns of distribution of DC+ cells (white arrows) in SGP (inset) and in the inner layers of the lateral tectum; (**E**) large constitutive niches in MS and SO (bounded by white dotted line) and single large bipolar DC+ cells (blue arrows) in SGC (inset) in the lateral tectum; (**F**) a dense layer of intensely labeled DC+ cells (white arrows) in SGP (inset) in the lateral tectum and migrating weakly labeled cells (blue arrows); (**G**) intensely labeled DC+ cells (white arrows) in SGP (blue inset) and DC+ RG (red arrowheads) in SM (black inset) in the dorsolateral tectum; (**H**) a heterogeneous population (inset) of DC+ cells (blue arrows) in the *torus longitudinalis* (TL) and RG fibers (red arrowheads); (**I**) intertoral commissures (CT) and a heterogeneous population of polygonal DC+ cells (inset) (other designations as in (**H**)). Scale bars: (**A**) 200 µm and (**B**–**I**) 100 µm. (**J**) A comparative distribution of DC+ cells (M ± SD) in the dorsal and lateral tectum; (**K**) a comparative distribution of DC+ neuroblasts and DC+ RG in the tectum with significant intergroup differences in the number of cells in the lateral and dorsal tectum, ## (*p* < 0.01) (*n* = 5 in each group), GS+ NE and GS+ RG in the tectum with significant intergroup differences in the number of cells in lateral tectum, # (*p* < 0.05) (*n* = 5 in each group), one-way ANOVA; (**L**) a comparative distribution of DC+ and GS+ cells in the tectum with significant intergroup differences in the number of cells in the lateral tectum, ## (*p* < 0.01) (*n* = 5 in each group), and in the dorsal tectum, # (*p* < 0.05) (*n* = 5 in each group), one-way ANOVA.

**Figure 5 ijms-23-01188-f005:**
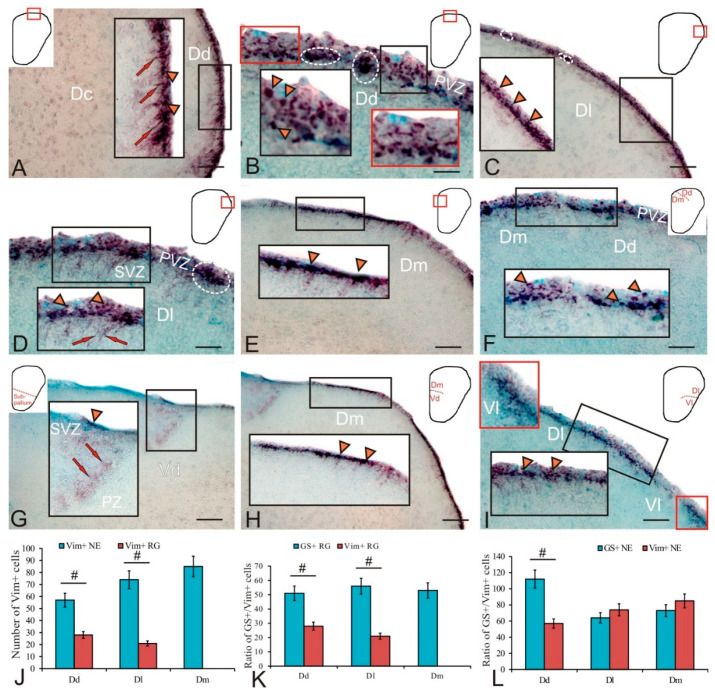
Vimentin (Vim) in the pallial and subpallial regions of the telencephalon in trout, *Oncorhynchus mykiss*: (**A**) immunolabeling of vimentin in PVZ (inset), Vim+ NE cells (orange arrowheads), and RG (red arrows) forming a monolayer; (**B**) Vim in Dd of pallium in small NE intensely labeled cells (insets) and constitutive clusters in the basal part of the PVZ (white dotted ovals), Vim+ NE cells (orange arrowheads); (**C**) Vim+ cells (orange arrowheads) in extensive clusters (inset) in Dl forming small dense groups (white dashed ovals); (**D**) Dl sectors (inset) containing RG (red arrows) (other designations, as in (**B**)); (**E**) Vim+ NE type cells in PVZ (inset), forming a monolayer in Dm; (**F**) a border zone between Dd and Dm (inset), with a lower density of Vim+ NE cells; (**G**) low Vim immunopositivity in the dorsal subpallium of Vd (inset), weakly labeled NE cells (red arrows), and Vim+ granules in the PVZ (orange arrowhead); (**H**) border between Dm and Vd and labeled NE cells (arrowheads) in Dm (inset); (**I**) border between Dl and Vl, clusters of Vim + cells in Dl (black inset), and a single aggregation of Vim+ cells of NE type (red inset). Scale bars: (**A**,**C**,**E**,**G**–**I**) 100 µm and (**B**,**D**,**F**) 50 µm. (**J**) A comparative distribution of Vim+ RG and NE cells in trout pallium, with significant intergroup differences in the number of cells in Dd and Dl, # (*p* < 0.05) (*n* = 5 in each group), one-way ANOVA; (**K**) a comparative distribution of Vim+ RG and GS+ RG in trout pallium, significant intergroup differences in the number of cells in Dd and Dl, # (*p* < 0.05) (*n* = 5 in each group), one-way ANOVA; (**L**) a comparative distribution of Vim+ NE and GS+ NE cells in trout pallium, significant intergroup differences in the number of cells in Dd, # (*p* < 0.05) (*n* = 5 in each group), one-way ANOVA.

**Figure 6 ijms-23-01188-f006:**
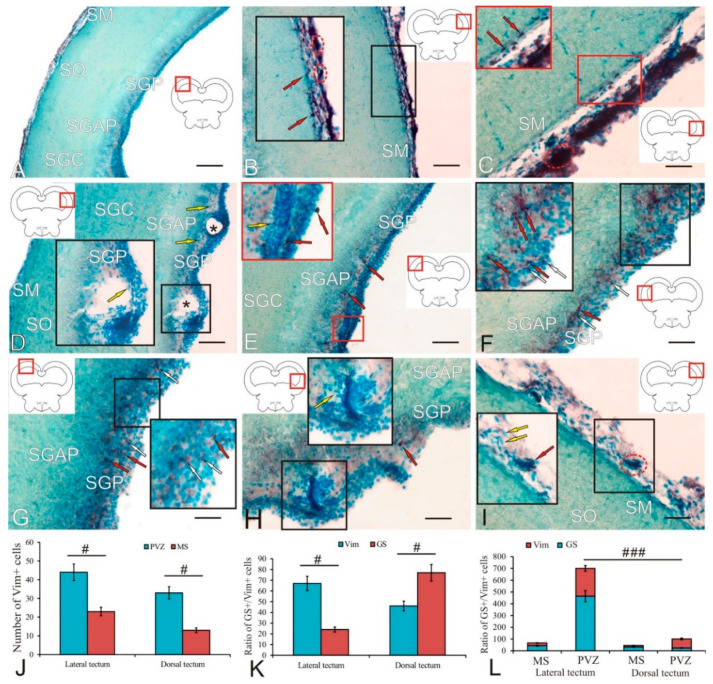
Vimentin (Vim) in the optic tectum of trout, *Oncorhynchus mykiss*: (**A**) a general view of distribution of vimentin in the tectum; (**B**) Vim+ cells (red arrows) and their clusters (in red dotted oval) in the ependyma and SM (inset) of the lateral tectum; (**C**) elongated intensely or moderately labeled Vim+ migrating cells (inset) in SM; (**D**) immunonegative cavities (asterisks) corresponding to lumens of large vessels (inset) in SGP and elongated Vim− cells (yellow arrows); (**E**) single Vim+ cells (red arrows) in SGP of the dorsolateral tectum (inset), Vim− cells (yellow arrows); (**F**) lateral thickening of SGP (inset) with a diffuse distribution of small intensely labeled Vim+ cells (red arrows) and larger weakly labeled cells (white arrows); (**G**) large moderately (red arrows) or weakly (white arrows) labeled Vim+ SGP cells (inset) of the dorsal tectum; (**H**) single Vim+ NE type cells (red arrow) and complex morphological patterns (inset) of resident and migrating immunonegative cells (yellow arrow) in the lateral proliferative zone of the tectum; (**I**) immunonegative cells (yellow arrows), a dense conglomerate of Vim+ cells (red dotted oval) in the ependyma (inset) of the dorsolateral tectum, and Vim+ cells (red arrow). Scale bars: (**A**) 200 µm, (**B**,**D**,**E**) 100 µm, and (**C**,**F**–**I**) 50 µm. (**J**) A comparative distribution of Vim+ cells in the lateral and dorsal tectum with significant intergroup differences in the number of cells in the PVZ and SM of the lateral and dorsal tectum, # (*p* < 0.05) (*n* = 5 in each group), one-way ANOVA; (**K**) a comparative distribution of Vim+ and GS+ cells in the lateral and dorsal tectum with significant intergroup differences in the number of cells in the PVZ and SM of the lateral and dorsal tectum, # (*p* < 0.05) (*n* = 5 in each group), one-way ANOVA; (**L**) a comparative distribution of Vim+ and GS+ cells in the lateral and dorsal tectum with significant intergroup differences in the number of cells in the PVZ of the lateral and dorsal tectum, ### (*p* < 0.001) (*n* = 5 in each group), one-way ANOVA.

**Figure 7 ijms-23-01188-f007:**
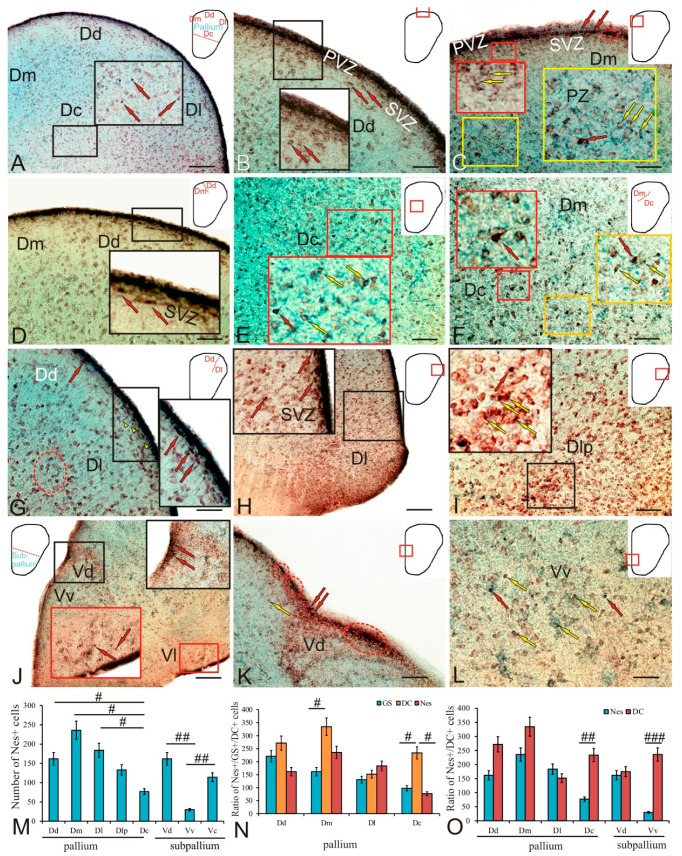
Nestin in the pallial and subpallial regions of the telencephalon in trout, *Oncorhynchus mykiss*: (**A**) a general view of nestin immunolabeling in the pallium and Nes+ cells (red arrows) of the central pallial zone Dc (inset); (**B**) intensely and moderately labeled Nes+ cells (inset) in Dd; (**C**) intensely labeled Nes+ cells in the PVZ (in white dotted oval), cells with the cytoplasmic localization of Nes (yellow arrows) in the SVZ (red inset), and Nes+ cells of the mixed type in the VZ (yellow inset) in Dm; (**D**) in the SVZ, horizontal cells with intense Nes+ labeling (red arrows) at the border between Dd and Dm (inset); (**E**) in Dc, moderately (yellow arrows) and intensely (red arrows) labeled Nes+ cells (inset); (**F**) polygonal moderately labeled (yellow inset) and large intensely labeled cells with processes (red arrow in inset) at the border between Dm and Dc; (**G**) at the border between Dd and Dl (inset), RG patterns (yellow arrowheads), densely labeled cells (red arrows), and cells with granular cytoplasmic localization of nestin (in red dotted oval); (**H**) a high density of distribution of Nes+ cells (red arrows) in Dl (inset); (**I**) a maximum density of distribution of Nes+ cells (red arrow) and granules (yellow arrows) in the posterolateral area of the pallium Dlp (inset); (**J**) Nes+ cells (red arrows) in the dorsal (Vd) (black inset), ventral (Vv), and lateral (Vl) (red inset) zones of the subpallium; (**K**) Nes+ cells in Vd, in the basal part of the PVZ, intensely labeled (in red dotted ovals) and extensive constitutive clusters of Nes+ cells (red arrows), and weakly labeled cells (yellow arrow); (**L**) large Nes+ cells in Vv (designations as in K). Scale bars: (**A**,**H**,**J**) 200 µm and (**B**–**G**,**I**,**K**,**L**) 100 µm. (**M**) A comparative distribution of Nes+ cells in the pallium and subpallium with significant intergroup differences in the number of cells in the pallium, # (*p* < 0.05) (*n* = 5 in each group), and subpallium, ## (*p* < 0.01) (*n* = 5 in each group), one-way ANOVA; (**N**) a comparative distribution of Nes+, DC+, and GS+ cells in the pallial region with significant intergroup differences in the number of cells in Dc and Dm, # (*p* < 0.05) (*n* = 5 in each group), two-way ANOVA; (**O**) a comparative distribution of Nes+ and DC+ cells of the NE type in the pallium and subpallium with significant intergroup differences in the number of Nes+ and DC+ cells in Dc of pallium, ## (*p* < 0.01) (*n* = 5 in each group) and Vv of subpallium, ### (*p* < 0.001) (*n* = 5 in each group), one-way ANOVA.

**Figure 8 ijms-23-01188-f008:**
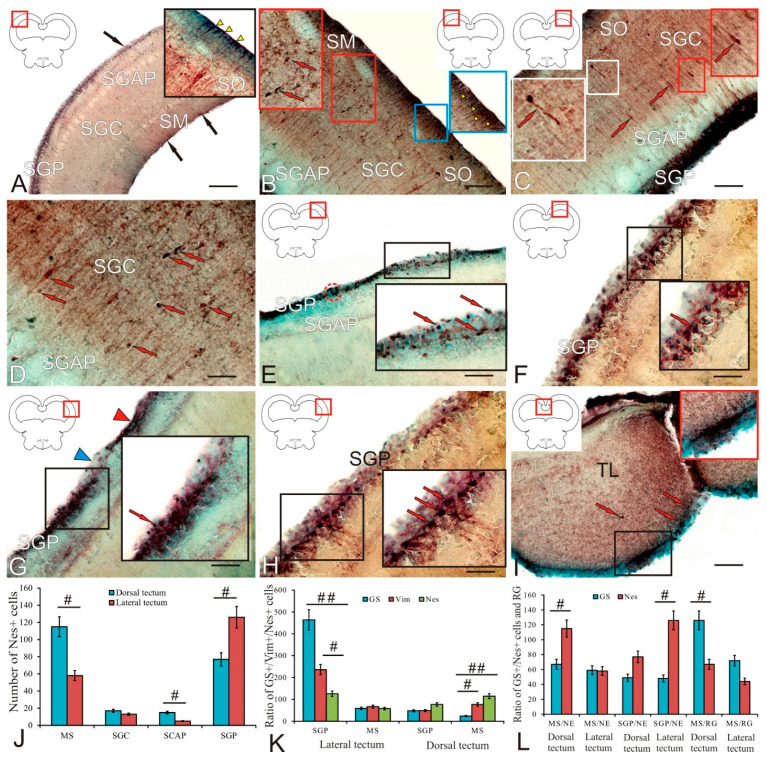
Nestin in the optic tectum of trout, *Oncorhynchus mykiss*: (**A**) a general view of nestin immunolabeling in the dorsolateral tectum of Nes+ NE type cells (black arrows) and Nes+ RG (yellow arrowheads); (**B**) Nes+ cells in SM (blue box) and moderately labeled Nes+ cells (red arrows) in SGC (red box); (**C**) intensely labeled cells (red arrows) with processes (white inset) in SGC and elongated bipolar cells (red inset) in SGAP; (**D**) different types of Nes+ cells (red arrows) in SGC; (**E**) local clusters of Nes+ cells (red dotted oval) and diffuse clusters (inset) of intensely labeled cells (red arrows) in SGP of the dorsolateral tectum; (**F**) a diffuse pattern of distribution of Nes+ cells in SGP (inset) in the dorsal tectum; (**G**) extensive densely labeled clusters of Nes+ cells (red arrowheads) in SGP (inset, single Nes+ cells indicated by red arrows), alternating with areas that lack immunopositive cells (blue arrowhead); (**H**) intensely labeled cells (red arrows) located diffusely in the basal part of SGP (inset); (**I**) Nes+ cells (red arrows) and clusters (inset) in the longitudinal torus (TL). Scale bars: (**A**) 200 µm, (**B**,**C**,**E**,**G**,**I**) 100 µm, and (**D**,**F**,**H**) 50 µm. (**J**) A comparative distribution of Nes+ cells in the dorsal and lateral tectum with significant intergroup differences in the number of cells in SM, SGAP, and SGP of the tectum, # (*p* < 0.05) (*n* = 5 in each group), one-way ANOVA; (**K**) a comparative distribution of GS+, Vim+, and Nes+ cells in different layers of the dorsal and lateral tectum with significant intergroup differences in the number of GS+ and Nes+ cells, ## (*p* < 0.01), Vim+ and Nes+ cells, # (*p* < 0.05) (*n* = 5 in each group) in SGP of lateral tectum and GS+ and Vim+ cells, # (*p* < 0.05), GS+ and Nes+ cells, ## (*p* < 0.01) (*n* = 5 in each group), in SM of dorsal tectum, two-way ANOVA; (**L**) a comparative distribution of GS+ and Nes+ cells of RG and NE types in SM and SGP of the dorsal and lateral tectum with significant intergroup differences in the number of GS+ and Nes+ NE cells in SM, # (*p* < 0.05), of dorsal tectum (*n* = 5 in each group), GS+ and Nes+ NE cells in SGP # (*p* < 0.05) of lateral tectum (*n* = 5 in each group), GS+ and Nes+ RG in SM, # (*p* < 0.05) of dorsal tectum (*n* = 5 in each group), one-way ANOVA.

**Table 1 ijms-23-01188-t001:** Morphometric and densitometric parameters (M ± SD) of glutamine-synthetase-immunopositive cells in the telencephalon and optic tectum of trout, *Oncorhynchus mykiss*.

Brain Area	Type of Cells, Brain Localization	Size of Cells, μm	Intensity of Labeling
**Telencephalon**			
*Pallium* Dorsal Dd	Undiffer. (PVZ) Elong. (PVZ, SVZ) Radial glia (PVZ)	5.5 ± 0.3/4.2 ± 0.7 7.0 ± 0.6/4.7 ± 0.7 9.4 ± 0.7/6.0 ± 0.6	+++ +++/++ +++
Medial Dm	Undiffer. (PVZ) Elongated (PVZ, PZ) Radial glia (PVZ)	4.9 ± 0.6/3.6 ± 0.8 6.7 ± 0.5/4.6 ± 0.8 8.8 ± 0.7/5.3 ± 0.9	+++ +++/++ +++
Lateral Dl	Undiffer. (PVZ) Elong. (PVZ, SVZ) Radial glia (PVZ)	5.2 ± 0.4/3.7 ± 0.5 7.2 ± 0.6/4.7 ± 0.8 8.4 ± 0.7/6.9 ± 1.5	+++ +++/++ +++
Central Dc	Undiffer. (PZ) Elongated (PZ)	5.2 ± 0.6/3.6 ± 0.4 7.0 ± 0.6/4.7 ± 0.7	+++/++ ++
*Subpallium* Dorsal Vd	Undiffer. (PVZ) Elongated (PVZ)	6.0 ± 0.6/5.3 ± 0.3 7.8 ± 0.4/5.4 ± 0.8	+++ +++/++
Ventral Vv	Elongated 1 (PVZ) Elongated 2 (PVZ)	7.5 ± 0.3/5.6 ± 0.6 8.9 ± 0.5/5.3 ± 0.4	+++ +++/++
Lateral Vl	Undiffer. (PVZ) Elongated (PVZ)	4.9 ± 0.5/3.6 ± 0.5 6.1 ± 0.3/4.0 ± 0.8	+++ +++/++
** *Tectum opticum* **			
Dorsal part *Stratum marginale* SM	Undifferentiated Elongated Radial glia 1 Radial glia 2	4.9 ± 0.1/3.1 ± 0.5 7.2 ± 0.7/5.7 ± 1.6 9.0 ± 0.4/5.5 ± 0.4 10.7 ± 0.4/6.0 ± 0.2	+++ +++ +++ +++
*Stratum griseum periventriculare* SGP	Elongated Radial glia 1	7.1 ± 0.5/4.6 ± 0.8 8.8 ± 0.5/5.2 ± 0.7	+++ +++
Lateral part *Stratum marginale* SM	Undifferentiated Elongated Radial glia 1	5.6 ± 0.3/3.9 ± 0.6 6.7 ± 0.6/4.3 ± 0.6 8.9 ± 0.4/5.3 ± 0.4	+++ +++ +++
*Stratum griseum periventriculare* SGP	Undifferentiated Elongated Radial glia 1 Radial glia 2	5.4 ± 0.2/4.3 ± 0.6 6.8 ± 0.5/4.6 ± 0.8 8.9 ± 0.6/5.7 ± 0.7 11.9 ± 0.3/5.1 ± 0.2	+++ +++ +++ +++

Optical density (OD) in immunopositive cells was categorized into high (180–130 UOD, designated as +++) and moderate (130–80 UOD, corresponding to ++); the initial OD value was measured on the control mounts. Values before slash are for the greater diameters of the cell body; after slash, for the lesser diameter.

**Table 2 ijms-23-01188-t002:** Morphometric and densitometric parameters (M ± SD) of doublecortin-immunopositive cells in the telencephalon and optic tectum of trout, *Oncorhynchus mykiss*.

Brain Area	Type of Cells, Brain Localization	Size of Cells, µm	Intensity of Labeling
**Telencephalon**			
*Pallium* Dorsal Dd	Undifferentiated (PVZ) Elongated (PVZ, SVZ) Oval (SVZ, PZ) Polygonal 1(PZ) Polygonal 2(PZ)	4.2 ± 0.4/3.1 ± 0.3 5.6 ± 0.8/4.0 ± 0.6 9.2 ± 0.9/8.2 ± 0.6 10.9 ± 0.9/8.6 ± 1.4 12.6 ± 0.4/9.4 ± 1.1	+++ +++/++ +++/++ ++ ++
Medial Dm	Undifferentiated (PVZ) Elongated (PVZ, SVZ) Oval (SVZ, PZ) Polygonal 1(PZ)	4.2 ± 0.2/3.1 ± 0.2 5.8 ± 0.6/4.0 ± 0.7 8.2 ± 0.4/5.8 ± 0.4 9.7 ± 0.3/5.1 ± 0.2	+++ +++/++ ++ ++
Lateral Dl	Elongated (PVZ, SVZ) Oval (SVZ, PZ) Polygonal 1(PZ) Polygonal 2(PZ)	5.3 ± 0.4/3.7 ± 0.4 9.3 ± 0.5/6.1 ± 0.7 11.1 ± 0.6/8.0 ± 1.6 12.7 ± 0.6/8.9 ± 0.8	+++ +++/++ +++ +++/++
Central Dc	Oval (SVZ, PZ) Polygonal 1(PZ) Polygonal 2(PZ)	10.1 ± 0.6/7.7 ± 1.2 11.8 ± 0.4/8.2 ± 0.6 13.7 ± 0.6/8.8 ± 1.4	+++/++ ++ ++
*Subpallium* Dorsal Vd	Undifferentiated (PVZ) Elongated (PVZ) Oval (SVZ, PZ)	4.6 ± 0.3/3.2 ± 0.4 5.6 ± 0.5/4.4 ± 0.7 7.5 ± 0.4/5.5 ± 0.9	+++ +++/++ +++/++
Ventral Vv	Oval (SVZ, PZ) Elongated 1 (SVZ, PZ) Elongated 2 (PZ)	5.6 ± 0.5/4.4 ± 0.7 7.5 ± 0.3/5.6 ± 0.6 8.9 ± 0.5/5.3 ± 0.4	+++ +++/++ ++
Central Vc	Elongated (PZ) Elongated 1 (PZ) Oval (PZ)	4.6 ± 0.5/3.4 ± 03 5.6 ± 0.5/4.4 ± 0.7 7.5 ± 0.4/5.5 ± 0.9	+++/++ ++ +++/++
Lateral Vl	Undifferentiated (PVZ) Elongated 1(SVZ, PVZ) Oval (SVZ, PZ) Polygonal 1(PZ)	4.5 ± 0.2/3.8 ± 0.3 5.9 ± 0.6/4.7 ± 0.8 7.7 ± 0.5/5.7 ± 1.4 10.2 ± 0.6/6.0 ± 0.2	+++ +++/++ ++ ++
** *Tectum Opticum* **			
*Stratum marginale* SM	Undifferentiated (PVZ) Elongated (PVZ) Radial glia 1(PVZ)	4.8 ± 0.5/3.5 ± 0.4 7.0 ± 0.7/4.4 ± 0.5 9.2 ± 0.8/5.5 ± 0.2	+++ +++ +++
*Stratum opticum* SO	Elongated (SVZ, PZ) Polygonal (PZ)	9.0 ± 0.7/5.7 ± 0.4 11.8 ± 2.0/6.4 ± 2.4	+++ +++
*Stratum griseumcentrale* SGC	Undifferentiated (PZ) Elongated (PZ) Oval (PZ) Polygonal 1(PZ) Polygonal 2(PZ)	5.6 ± 0.3/4.2 ± 1.0 9.4 ± 0.9/6.4 ± 0.4 11.4 ± 0.4/9.0 ± 2.0 13.2 ± 0.6/7.7 ± 0.4 15.3 ± 0.3/6.3 ± 1.2	+++ +++/++ ++ ++ ++
*Stratum griseum et al. bum periventriculare* SGAP	Undifferentiated (SVZ) Elongated (SVZ, PZ) Oval (SVZ, PZ)	6.4 ± 0.4/4.6 ± 0.4 9.3 ± 0.4/5.2 ± 0.5 10.8 ± 0.2/6.4 ± 0.6	+++ ++ ++
*Stratum griseum* *periventriculare* SGP	Undifferentiated (PVZ) Elongated (PVZ) Oval (PVZ)	5.3 ± 0.4/4.0 ± 0.7 6.7 ± 0.6/4.5 ± 0.8 8.7 ± 0.6/6.3 ± 0.3	+++ +++ +++
*Torus longitudinalis* TL	Undifferentiated (PVZ) Elongated 1(PVZ) Elongated 2 (PVZ) Polygonal (PZ)	4.7 ± 0.6/3.0 ± 0.5 7.0 ± 0.6/5.5 ± 1.4 8.7 ± 0.6/7.0 ± 1.6 11.2 ± 0.4/6.7 ± 1.3	+++ +++/++ ++ ++

Optical density (OD) in immunopositive cells was categorized into high (180–130 UOD, corresponding to +++) and moderate (130–80 UOD, corresponding to ++); the initial OD value was measured on the control mounts. Values before slash are for the greater diameters of the cell body; after slash, for the lesser diameter.

**Table 3 ijms-23-01188-t003:** Morphometric and densitometric parameters (M ± SD) of vimentin-immunopositive cells in the telencephalon and optic tectum of trout, *Oncorhynchus mykiss*.

Brain Area	Type of Cells, Brain Localization	Size of Cells, µm	Intensity of Labeling
**Telencephalon**			
*Pallium* Dorsal Dd	Undiffer. (PVZ) Elongated (PVZ) Radial glia (PVZ)	4.0 ± 0.4/2.8 ± 0.6 5.7 ± 0.6/4.1 ± 1.1 8.4 ± 0.4/6.9 ± 0.9	+++ +++/++ +++
Lateral Dl	Undiffer. (PVZ) Elongated (PVZ, PZ) Radial glia (PVZ)	3.9 ± 0.5/2.8 ± 0.5 6.2 ± 0.7/4.0 ± 0.6 8.3 ± 0.5/5.3 ± 0.5	+++ +++/++ +++
Medial Dm	Undiffer. (PVZ) Elongated (PVZ)	3.5 ± 0.3/2.5 ± 0.3 4.9 ± 0.4/3.4 ± 0.6	+++ +++
Central Dc	-	-	-
*Subpallium* Dorsal Vd	Undiffer. (PVZ) Elongated 1(SVZ) Elongated 2 (PZ)	6.0 ± 0.6/5.3 ± 0.3 7.8 ± 0.4/5.4 ± 0.8	+++ + +
Ventral Vv	-	-	-
Dl/Vl	Undiffer. (PVZ) Elongated (PVZ)	3.4 ± 0.3/2.6 ± 0.6 4.7 ± 0.5/3.3 ± 0.5	+++ +++
* **Tectum Opticum** *			
*Stratum marginale* SM	Undiffer. 1 (SM) Undiffer. 2 (SM) Elongated (SM) Elongated 1 (SM) Elongated 2 (SM)	3.3 ± 0.5/2.3 ± 0.3 5.8 ± 1.0/4.0 ± 1.1 8.0 ± 0.6/6.2 ± 1.1 9.9 ± 0.3/5.6 ± 0.3 13.6 ± 0.4/8.4 ± 2.9	+++ +++ +++ +++/++ +++/++
*Stratum marginale* Neurogenic nishes NN	ependyma ependyma ependyma ependyma ependyma	12.2 ± 0.8/8.7 ± 2.0 16.5 ± 0.7/‘11.3 ± 0.7 20.8 ± 0.6/19.0 ± 9.1 24.8 ± 1.4/15.3 ± 5.3 28.5 ± 1.6/20.3 ± 8.9	+++ +++ +++ +++ +++
*Stratum griseum periventriculare* SGP	Undiffer. 1(SGP) Undiffer. 2 (SGP) Elongated (SGP) Elongated 1 (SGP) Elongated 2 (SGP)	4.3 ± 1.2/2.7 ± 0.5 7.0 ± 0.3/5.7 ± 0.9 9.0 ± 0.3/6.7 ± 0.7 10.8 ± 0.5/7.5 ± 0.9 13.3 ± 1.4/9.4 ± 1.9	+++ +++ +++ +++/++ +++/++

Optical density (OD) in immunopositive cells was categorized into high (180–130 UOD, corresponding to +++), moderate (130–80 UOD, corresponding to ++), weak (80–40 UOD, corresponding to +), and low (less than 40 UOD, corresponding to −); the initial OD value was measured on the control mounts. Values before slash are for the greater diameters of the cell body; after slash, for the lesser diameter.

**Table 4 ijms-23-01188-t004:** Morphometric and densitometric parameters (M ± SD) of nestin-immunopositive cells in the telencephalon and optic tectum of trout, *Oncorhynchus mykiss*.

Brain Area	Type of Cells, Brain Localization	Size of Cells, µm	Intensity of Labeling
**Telencephalon**			
*Pallium* Dorsal Dd	Undiffer. (PVZ) Elong. (PVZ, SVZ) Oval (SVZ, PZ)	5.0 ± 0.5/3.3 ± 0.2 6.7 ± 0.5/4.1 ± 0.6 9.0 ± 0.7/4.2 ± 0.6	+++ +++/++ +++/++
Medial Dm	Undiffer. (PVZ) Elong. (PVZ, SVZ) Oval (SVZ, PZ)	5.7 ± 0.3/4.6 ± 0.9 7.2 ± 0.4/4.0 ± 0.8 8.8 ± 0.5/6.2 ± 1.3	+++ +++/++ +++/++
Dm/Dc	Elong. (PZ) Oval (PZ) Polygonal 1 (PZ) Polygonal 2 (PZ)	7.7 ± 0.4/6.2 ± 0.5 10.2 ± 0.9/5.9 ± 0.3 11.9 ± 0.8/7.7 ± 1.6 17.8 ± 1.6/10.2 ± 0.8	+++ +++/++ +++/++ +++/++
Central Dc	Oval (SVZ, PZ) Polygonal (PZ)	11.1 ± 0.4/8.9 ± 1.0 12.8 ± 0.3/8.8 ± 0.9	+++/++ ++
Lateral Dl	Undiffer. (PVZ) Elong. (PVZ, SVZ) Oval (SVZ, PZ) Polygonal (SVZ)	5.2 ± 0.5/3.9 ± 0.5 7.0 ± 0.5/4.4 ± 0.7 9.3 ± 0.7/5.4 ± 0.4 11.3 ± 0.6/3.7 ± 0.8	+++ +++/++ +++ +++/++
Lateral posterior zone Dlp	Elong. (PVZ, SVZ) Oval (SVZ, PZ) Polygonal 1 (PZ) Polygonal 2 (PZ)	7.3 ± 0.4/6.5 ± 1.3 9.8 ± 0.5/7.0 ± 0.3 11.9 ± 0.6/6.6 ± 0.9 11.6 ± 0.4/7.8 ± 1.0	+++ +++/++ +++ +++/++
*Subpallium* Dorsal Vd	Undiffer. (PVZ) Elongated (PVZ)	5.0 ± 0.4/3.6 ± 0.4 7.0 ± 0.4/4.5 ± 0.3	+++ +++/++
Ventral Vv	Oval (SVZ, PZ) Elong. 1 (SVZ, PZ) Elongated 2 (PZ)	9.0 ± 0.6/8.0 ± 1.2 11.1 ± 0.6/8.3 ± 1.6 13.4 ± 0.4/8.7 ± 2.3	+++ +++/++ ++
Lateral Vl	Undiffer. (PVZ) Elongated 1(PVZ, PZ) Oval (PVZ) Polygonal (PZ)	4.5 ± 0.4/3.8 ± 0.5 5.9 ± 0.5/4.5 ± 0.7 7.8 ± 0.6/5.5 ± 1.4 10.2 ± 0.6/6.0 ± 0.2	+++ +++/++ ++ ++
** *Tectum opticum* **			
*Stratum marginale* SM	Undiffer. (PVZ) Elongated (PVZ) Radial glia 1(PVZ)	4.3 ± 0.5/3.2 ± 0.4 5.8 ± 0.5/3.8 ± 0.7 8.0 ± 1.4/3.5 ± 0.7	+++ +++ +++
*Stratum griseum centrale* SGC	Elongated (PZ) Oval (PZ) Polygonal (PZ)	8.5 ± 0.5/5.3 ± 1.1 11.0 ± 1.0/6.3 ± 1.3 26.2 ± 2.5/7.3 ± 1.3	+++/++ ++ ++
*Stratum griseum periventriculare* SGP	Undiffer. (PVZ) Elongated (PVZ)	4.8 ± 0.6/3.5 ± 0.6 6.7 ± 0.6/4.5 ± 0.9	+++ +++
*Torus longitudinalis* TL	Undiffer. (PVZ) Elongated 1(PVZ) Elongated 2 (PVZ)	5.5 ± 0.4/3.4 ± 0.6 6.3 ± 0.2/3.9 ± 0.8 8.3 ± 0.6/3.3 ± 0.4	+++ +++/++ ++
Parenchymal cells PC	Undiffer. (PVZ) Elongated (PVZ)	4.8 ± 0.5/3.3 ± 0.3 6.3 ± 0.4/3.7 ± 0.4	+++ +++/++

Optical density (OD) in immunopositive cells was categorized into high (180–130 UOD, corresponding to +++) and moderate (130–80 UOD, corresponding to ++); the initial OD value was measured on the control mounts. Values before slash are for the greater diameters of the cell body; after slash, for the lesser diameter.

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
