# Peer review of "Molecular Markers of Adult Neurogenesis in the Telencephalon and Tectum of Rainbow Trout, Oncorhynchus mykiss"

_ijms, 2022, doi:10.3390/ijms23031188_

Round 1

Reviewer 1 Report

Pushchina and colleagues (ID IJMS: ijms-1548387-peer-review-v1) investigated the distribution of several molecular markers in the telencephalon and tectum on 3-year-old rainbow trout, Oncorkynchus mykiss, aiming at identifying which cells are relevant for the development of neurogenesis processes in the adult stage.

The topic of neurogenesis in teleost fish has recently seen an increasing interest from translational research. The aim of the present study sounds interesting and relevant for understanding the continuous processes of neurogenesis in the adult stage. The introduction is well written and reported relevant references for understanding the topic of this study. The methods are correctly reported and the results are discussed.  I consider the work to be suitable for publication.

I have a few minor comments that I would author consider for improving their work.

Abstract

I would suggest the authors reconsider the detail in their abstract. At the moment, the abstract fully describes their results. There is no mention of the reasons that brought the authors to conduct this study. In addition, the present Journal is not focused only on fish, so authors should help readers in understanding the research topic, the adopted methodology, and their findings. It would be great if the authors could elaborate a bit more on the purpose of their study, what was done and what was found.

Introduction

The second paragraph sounds untied with the first paragraph. I would suggest connecting the first two paragraphs with a brief description of the importance of neurogenesis study in teleost fish.

Lines 50-55. I would suggest reordering this paragraph. Firstly, the authors reported descriptive information of fish body weights and size. Then, they started focusing on the brain volume. Before mentioning that various aspects have been found to be relevant for constitutive neurogenesis (lines 50-52), I would shortly describe this process and which cells are relevant for (as reported from 52-55). After this description, I would describe a bit more which external factors have an impact on this process (line 50-52).

Results

Tables 1,2, and 3. A short description of '+' meanings for the intensity of labeling should be mentioned in the caption.

Discussion

Lines 761: I would avoid the term “evolutionarily advanced” vertebrates. How primates should be considered more advanced with respect to fish or even invertebrates? I would suggest removing the last part of this sentence from “ranging from the most” to the end.

Author Response

Dear Reviewer, we thank you for your great work in reviewing our article. Thanks to your valuable comments, we have been able to significantly improve the quality of our work. Below is a list of specific changes that have been made to work in line with your comments:

Abstract

I would suggest the authors reconsider the detail in their abstract. At the moment, the abstract fully describes their results. There is no mention of the reasons that brought the authors to conduct this study. In addition, the present Journal is not focused only on fish, so authors should help readers in understanding the research topic, the adopted methodology, and their findings. It would be great if the authors could elaborate a bit more on the purpose of their study, what was done and what was found.

 The details of the abstract have been revised in accordance with the recommendations; the purpose of the work is detailed in the final part.

Introduction

The second paragraph sounds untied with the first paragraph. I would suggest connecting the first two paragraphs with a brief description of the importance of neurogenesis study in teleost fish.

Lines 50-55. I would suggest reordering this paragraph. Firstly, the authors reported descriptive information of fish body weights and size. Then, they started focusing on the brain volume. Before mentioning that various aspects have been found to be relevant for constitutive neurogenesis (lines 50-52), I would shortly describe this process and which cells are relevant for (as reported from 52-55). After this description, I would describe a bit more which external factors have an impact on this process (line 50-52).

Appropriate corrections have been made.

Results

Tables 1,2, and 3. A short description of '+' meanings for the intensity of labeling should be mentioned in the caption.

 Descriptions added

Discussion

Lines 761: I would avoid the term “evolutionarily advanced” vertebrates. How primates should be considered more advanced with respect to fish or even invertebrates? I would suggest removing the last part of this sentence from “ranging from the most” to the end.

The recommendations have been implemented.

Reviewer 2 Report

Dear authors,

I report below my comments and suggestions related to your interesting work. In my opinion, the text appears in some points unclear and difficult to read for the following reasons.

-In the Introduction section, references should be given in support of the study of glutamine synthetase, doublecortin, vimentin and nestin.

-There are numerous abbreviations and acronyms in the work. A table summarizing all the acronyms would be useful for understanding the text.

-Abbreviations and acronyms must be defined the first time they are used. For example, the meaning of NSCs is not specified in Line 17.

-Acronyms do not always correspond to the initials of the words they refer to and this makes reading more difficult. For example, neuroepithelial neural stem cells are referred to as eNSCs and not nNSCs.

-Once the acronym has been defined, it is not necessary to define it again. For example, “neuroepithelial (NE)” is reported in both Line 58 and Line 61.

-Line 109. Replace “GS + cell” with “GS positive (GS +) cell”.

-Line 109: "the bodies of the" means "the cell bodies of the"?

-Line 144: “three types of small, moderately and intensely labeled cells” is an unclear sentence. “Small” refers to the size of the cells or to the fact that they have weak labeling? The three cell types are: small, moderately, and intensely labeled? Please rephrase the sentence.

-Line 165: “intensely labeled neuroepithelial cells”. If the acronym NE is defined, I would always use it. So, I would replace the phrase with "intensely labeled NE cells".

-Line 167: "In the SVZ, there were single moderately or weakly labeled cells." lease change to "Single moderately or weakly labeled cells were observed in the SVZ.".

-Line 202: "In the subpallial region of the telencephalon of juvenile chum salmon, GS was found". In the methods it is reported that "The study was carried out on 20 three-year-old rainbow trout Oncorhynchus mykiss" (line 1205). If only trout were used, why is there data for chum salmon? If the study carried out involves the use of chum salmon, this must be declared in the materials and methods section.

-Line 217: "(undifferentiated small and oval cells)" please replace with "(small and oval undifferentiated cells)"

-Line 228: the meaning of “type 1” and “type2” is not clear.

-In my opinion, the captions of the figures and the text are separate entities. Acronyms used in the text must be specified in the text the first time they are used. So, in line 265 you should specify the meaning of LOT.

-Lines 281-281: “periventricular zone (SGP)”. Why is the acronym SGP and not PVZ as defined in line 107?

-Line 283: “It contained both homogeneous areas containing NE cells” the term “contain” is repeated. Please rephrase the sentence avoiding repetition.

-Line 295: "while in the MS, on the contrary, the total number of GS + cells" change to "on the contrary in the MS the total number of GS + cells".

-Lines 316-317: “Dc-immunopositive” replace with “DC-immunopositive”. Likewise in line 318.

-Line 326: is “dominated” the more correct term? Does it mean that they were very numerous or the most numerous?

-Line 485: "Vim-positive", the acronym Vim has not been defined in the text.

-Line 528: "Weakly labeled diffusely organized cells" please replace with "Diffusely organized weakly labeled cells".

-Line 534: choose whether positivity should be indicated as "Vim-positive", "vim-immunopositive" or "vim +" and always use that expression in the text.

-Line 542: “Vim + and GS + RG in“ the term “cells” is missing. The same on line 544.

-Line 554: “NE type cells”. Sometimes it is written "NE type cells" and sometimes "NE cells". Please use the same expression or distinguish the different meanings of the two expressions.

-Lines 554-555: "above SM, neurogenic niches of various sizes were found in ependyma" replace with "neurogenic niches of various sizes were found in ependyma above SM".

-Lines 670-671; the pallium and subpallium zones " please replace with" pallial and subpallial zones "

-Line 685: "the pallium and subpallium of trout is shown in Fig. 7O.". If the work is based on trout (as stated in the title and in materials and methods), then "of trout" can be removed. The same in line 692 and in other parts of the text.

-Line 737 shows “Nes + NE type cells”. Line 753 shows “NE type Nes +”. Why are they written differently? Should both be “Nes + NE type cells” or “Nes + NE cells”, or do the two expressions have different meanings?

-Lines 80-81: I suggest changing the sentence "The analysis of RG growth using immunocytochemical markers of proliferation PCNA, stem cells Sox2, and neuronal differentiation doublecortin" to "The analysis of RG growth using the immunocytochemical markers of proliferation PCNA, stem cells Sox2 , and neuronal differentiation doublecortin "or" The analysis of RG growth using immunocytochemical markers of proliferation such as PCNA, stem cells Sox2, and neuronal differentiation doublecortin ".

Figures and captions

-The colours used for the arrows in the figures are sometimes not very visible. I suggest using deeper colours or larger arrows.

-If the arrow of a certain type and colour always points to the same cell in all the panels of the figure, then I suggest reporting at the bottom of the caption the list of all the arrows used and the cells to which they refer.

Figure 1:

-Line 117: “radial glia (blue arrows),“ the colour seems to me “light blue”. A more contrasted blue arrow would be better.

-Line 119: “intensely labeled GS cells” means “intensely labeled GS + cells”?

-Lines 122-123: what do the green arrows indicate?

-Line 124: the red dotted oval is not very visible in the figure

-Lines 125-127: the meaning of the black arrows is not indicated.

-Line 129: “on GS– cells” means “GS negative cells”?

-Lines 127-129. Panel G: the meaning of the white arrows is not indicated.

-Lines 129-131: the meaning of the white arrows is not indicated.

-Lines 133-134: what is the meaning of the green arrows?

Figure 2

-Line 232: “triangular arrows”. They are sometimes referred to as "arrowheads". Choose one of the two options and always use the one selected in the text. I would prefer "arrowheads".

Figure 3

-Lines 342-343: panel G. what do the black dotted ovals mean?

-In panels M, N, O, P the letters (M, N, O, P) are not very visible. I suggest writing them in black.

-In the graphs where the ratio is represented (panels R-T) and in the following figures, it would be better to express the ratio as a percentage.

Figure 5

-Lines 490-492: Panel B: what do orange arrowheads indicate?

-Does the blue colour observed in the insets correspond to arrowheads or histological staining?

-Lines 498: replace "(triangular arrowheads)" with "triangular arrow" or "arrowhead"

Figure 6

-Line 588: "visual tectum" means "optic tectum"?

-Line 593: "(yellow arrow);" replace with "(yellow arrows)"

-Lines 593-594: what does the yellow arrow indicate?

-Lines 599-601: panel I. What does the red arrow indicate?

Figure 7

-Line 636: "labeled Nes cells" replace with "labeled Nes + cells"

-Lines 635-636: What do the red arrows indicate?

-Line 636- Panel C- the red dotted oval in the figure is very little visible

-Line 638: “labeled with intensive Nes labeling” label is repeated. Please change the sentence

-Line 639: "intensively (red arrows)" replace with "intensively (red arrow)"

-Lines 639-641. Panel F. The meaning of the red arrow is not indicated

-Panel M. In panel G of Figure 4 the ordinate is "Number of DC + cells, n". In other panels, "n" is not reported. Please standardize by adding ", n" when necessary

Figure 8

-Line 714: "Nes cells" please replace with "Nes + cells"

-Lines 714-715. Panel C. What do the red arrows mean?

-Lines 717-719. Panel G. What does the red arrow indicate?

Tables

-In the caption of the tables instead of "expressing cells" I would replace with "positive cells".

-Line 902: “immunopositive Vim + RG cells” is redundant. I would replace with “Vim + RG cells”.

Methods

-Line 1208 Please indicate tank size and fish density

-Line 1208 Please indicate the origin of the water and its chemical and physical parameters

-Lines 1217-1220 It is suggested to modify the sentence "The animals were removed from the experiment and euthanized by the method of rapid decapitation. They were anesthetized in a 0.1% solution of ethyl-3-aminobenzoate methanesulfonate (MS222) (Sigma, St. Louis, Cat. # 1219 WXBC9102V MO, USA) for 10–15 min. " in "The animals were removed from the experiment, anesthetized in a 0.1% solution of ethyl-3-aminobenzoate methanesulfonate (MS222) (Sigma, St. Louis, Cat. # 1219 WXBC9102V MO, USA) for 10–15 min and euthanized by the method of rapid decapitation. "

-Lines 1226-1227 "Every third frontal section of the cerebellum was taken for the reaction." Are the sections kept floating or collected directly on the slide?

-Immunohistochemistry section. Since data deriving from densitometric analysis are presented, it is necessary to provide specific information on the time of incubation and development of the reactions.

-It seems to me that the experimental protocol is incomplete. Before incubation with the antibody, the sections are not permeabilized with Triton X-100 or Tween 20? Please provide all the necessary information to exactly replicate the experiment performed.

-Line 1233. What is meant by “sections incubated in situ”? Is the incubation performed on floating sections or on sections adhered to the slide?

-Line 1264 please change “(180–130 UOD, corresponding to +++)” to “(180–130 UOD, corresponding to +++ in tables n ...).

-Line 1296: “tectum parenchyma” mean “tectal parenchyma”?

Author Response

Dear Reviewer, we thank you for your great work in reviewing our article. Thanks to your valuable comments, we have been able to significantly improve the quality of our work. Below is a list of specific changes that have been made to work in line with your comments:

All changes in the text, captions to figures, tables are highlighted in yellow.

In accordance with your recommendations, the English language was revised in the article

 -In the Introduction section, references should be given in support of the study of glutamine synthetase, doublecortin, vimentin and nestin.

The recommendations have been implemented.

-There are numerous abbreviations and acronyms in the work. A table summarizing all the acronyms would be useful for understanding the text.

The list of abbreviations has been added to the article

-Abbreviations and acronyms must be defined the first time they are used. For example, the meaning of NSCs is not specified in Line 17.

The recommendations have been implemented

-Acronyms do not always correspond to the initials of the words they refer to and this makes reading more difficult. For example, neuroepithelial neural stem cells are referred to as eNSCs and not nNSCs.

corrected

-Once the acronym has been defined, it is not necessary to define it again. For example, “neuroepithelial (NE)” is reported in both Line 58 and Line 61.

-Line 109. Replace “GS + cell” with “GS positive (GS +) cell”.

-Line 109: "the bodies of the" means "the cell bodies of the"?

-Line 144: “three types of small, moderately and intensely labeled cells” is an unclear sentence. “Small” refers to the size of the cells or to the fact that they have weak labeling? The three cell types are: small, moderately, and intensely labeled? Please rephrase the sentence.

-Line 165: “intensely labeled neuroepithelial cells”. If the acronym NE is defined, I would always use it. So, I would replace the phrase with "intensely labeled NE cells".

-Line 167: "In the SVZ, there were single moderately or weakly labeled cells." lease change to "Single moderately or weakly labeled cells were observed in the SVZ.".

-Line 202: "In the subpallial region of the telencephalon of juvenile chum salmon, GS was found". In the methods it is reported that "The study was carried out on 20 three-year-old rainbow trout Oncorhynchus mykiss" (line 1205). If only trout were used, why is there data for chum salmon? If the study carried out involves the use of chum salmon, this must be declared in the materials and methods section.

-Line 217: "(undifferentiated small and oval cells)" please replace with "(small and oval undifferentiated cells)"

-Line 228: the meaning of “type 1” and “type2” is not clear.

-In my opinion, the captions of the figures and the text are separate entities. Acronyms used in the text must be specified in the text the first time they are used. So, in line 265 you should specify the meaning of LOT.

-Lines 281-281: “periventricular zone (SGP)”. Why is the acronym SGP and not PVZ as defined in line 107?

-Line 283: “It contained both homogeneous areas containing NE cells” the term “contain” is repeated. Please rephrase the sentence avoiding repetition.

-Line 295: "while in the MS, on the contrary, the total number of GS + cells" change to "on the contrary in the MS the total number of GS + cells".

-Lines 316-317: “Dc-immunopositive” replace with “DC-immunopositive”. Likewise in line 318.

-Line 326: is “dominated” the more correct term? Does it mean that they were very numerous or the most numerous?

-Line 485: "Vim-positive", the acronym Vim has not been defined in the text.

-Line 528: "Weakly labeled diffusely organized cells" please replace with "Diffusely organized weakly labeled cells".

-Line 534: choose whether positivity should be indicated as "Vim-positive", "vim-immunopositive" or "vim +" and always use that expression in the text.

-Line 542: “Vim + and GS + RG in“ the term “cells” is missing. The same on line 544.

-Line 554: “NE type cells”. Sometimes it is written "NE type cells" and sometimes "NE cells". Please use the same expression or distinguish the different meanings of the two expressions.

-Lines 554-555: "above SM, neurogenic niches of various sizes were found in ependyma" replace with "neurogenic niches of various sizes were found in ependyma above SM".

-Lines 670-671; the pallium and subpallium zones " please replace with" pallial and subpallial zones "

-Line 685: "the pallium and subpallium of trout is shown in Fig. 7O.". If the work is based on trout (as stated in the title and in materials and methods), then "of trout" can be removed. The same in line 692 and in other parts of the text.

-Line 737 shows “Nes + NE type cells”. Line 753 shows “NE type Nes +”. Why are they written differently? Should both be “Nes + NE type cells” or “Nes + NE cells”, or do the two expressions have different meanings?

-Lines 80-81: I suggest changing the sentence "The analysis of RG growth using immunocytochemical markers of proliferation PCNA, stem cells Sox2, and neuronal differentiation doublecortin" to "The analysis of RG growth using the immunocytochemical markers of proliferation PCNA, stem cells Sox2 , and neuronal differentiation doublecortin "or" The analysis of RG growth using immunocytochemical markers of proliferation such as PCNA, stem cells Sox2, and neuronal differentiation doublecortin ".

 All recommended corrections are included in the Results section

Figures and captions

-The colours used for the arrows in the figures are sometimes not very visible. I suggest using deeper colours or larger arrows.

-If the arrow of a certain type and colour always points to the same cell in all the panels of the figure, then I suggest reporting at the bottom of the caption the list of all the arrows used and the cells to which they refer.

 Figure 1:

-Line 117: “radial glia (blue arrows),“ the colour seems to me “light blue”. A more contrasted blue arrow would be better.

-Line 119: “intensely labeled GS cells” means “intensely labeled GS + cells”?

-Lines 122-123: what do the green arrows indicate?

-Line 124: the red dotted oval is not very visible in the figure

-Lines 125-127: the meaning of the black arrows is not indicated.

-Line 129: “on GS– cells” means “GS negative cells”?

-Lines 127-129. Panel G: the meaning of the white arrows is not indicated.

-Lines 129-131: the meaning of the white arrows is not indicated.

-Lines 133-134: what is the meaning of the green arrows?

 Figure 2

-Line 232: “triangular arrows”. They are sometimes referred to as "arrowheads". Choose one of the two options and always use the one selected in the text. I would prefer "arrowheads".

 Figure 3

-Lines 342-343: panel G. what do the black dotted ovals mean?

-In panels M, N, O, P the letters (M, N, O, P) are not very visible. I suggest writing them in black.

-In the graphs where the ratio is represented (panels R-T) and in the following figures, it would be better to express the ratio as a percentage.

 Figure 5

-Lines 490-492: Panel B: what do orange arrowheads indicate?

-Does the blue colour observed in the insets correspond to arrowheads or histological staining?

-Lines 498: replace "(triangular arrowheads)" with "triangular arrow" or "arrowhead"

Figure 6

-Line 588: "visual tectum" means "optic tectum"?

-Line 593: "(yellow arrow);" replace with "(yellow arrows)"

-Lines 593-594: what does the yellow arrow indicate?

-Lines 599-601: panel I. What does the red arrow indicate?

 Figure 7

-Line 636: "labeled Nes cells" replace with "labeled Nes + cells"

-Lines 635-636: What do the red arrows indicate?

-Line 636- Panel C- the red dotted oval in the figure is very little visible

-Line 638: “labeled with intensive Nes labeling” label is repeated. Please change the sentence

-Line 639: "intensively (red arrows)" replace with "intensively (red arrow)"

-Lines 639-641. Panel F. The meaning of the red arrow is not indicated

-Panel M. In panel G of Figure 4 the ordinate is "Number of DC + cells, n". In other panels, "n" is not reported. Please standardize by adding ", n" when necessary

Figure 8

-Line 714: "Nes cells" please replace with "Nes + cells"

-Lines 714-715. Panel C. What do the red arrows mean?

-Lines 717-719. Panel G. What does the red arrow indicate?

 All recommended corrections are included in the Figure captions section

  Tables

-In the caption of the tables instead of "expressing cells" I would replace with "positive cells".

-Line 902: “immunopositive Vim + RG cells” is redundant. I would replace with “Vim + RG cells”.

 All recommended corrections are included in the Tables

 Methods

 -Line 1208 Please indicate tank size and fish density

-Line 1208 Please indicate the origin of the water and its chemical and physical parameters

-Lines 1217-1220 It is suggested to modify the sentence "The animals were removed from the experiment and euthanized by the method of rapid decapitation. They were anesthetized in a 0.1% solution of ethyl-3-aminobenzoate methanesulfonate (MS222) (Sigma, St. Louis, Cat. # 1219 WXBC9102V MO, USA) for 10–15 min. " in "The animals were removed from the experiment, anesthetized in a 0.1% solution of ethyl-3-aminobenzoate methanesulfonate (MS222) (Sigma, St. Louis, Cat. # 1219 WXBC9102V MO, USA) for 10–15 min and euthanized by the method of rapid decapitation. "

-Lines 1226-1227 "Every third frontal section of the cerebellum was taken for the reaction." Are the sections kept floating or collected directly on the slide?

-Immunohistochemistry section. Since data deriving from densitometric analysis are presented, it is necessary to provide specific information on the time of incubation and development of the reactions.

-It seems to me that the experimental protocol is incomplete. Before incubation with the antibody, the sections are not permeabilized with Triton X-100 or Tween 20? Please provide all the necessary information to exactly replicate the experiment performed.

-Line 1233. What is meant by “sections incubated in situ”? Is the incubation performed on floating sections or on sections adhered to the slide?

-Line 1264 please change “(180–130 UOD, corresponding to +++)” to “(180–130 UOD, corresponding to +++ in tables n ...).

-Line 1296: “tectum parenchyma” mean “tectal parenchyma”?

 All recommended corrections are included in the Methods section
